# Toward Effective Deep Reinforcement Learning for 3D Robotic Manipulation: Multimodal End-to-End Reinforcement Learning from Visual and Proprioceptive Feedback

## Abstract

Sample-efficient reinforcement learning (RL) methods capable of learning directly from raw sensory data without the use of human-crafted representations would open up real-world applications in robotics and control. Recent advances in visual RL have shown that learning a latent representation together with existing RL algorithms closes the gap between state-based and image-based training. However, image-based training is still significantly sample-inefficient with respect to learning in 3D continuous control problems (for example, robotic manipulation) compared to state-based training. In this study, we propose an effective model-free off-policy RL method for 3D robotic manipulation that can be trained in an end-to-end manner from multimodal raw sensory data obtained from a vision camera and a robot's joint encoders. Most notably, our method is capable of learning a latent multimodal representation and a policy in an efficient, joint, and end-to-end manner from multimodal raw sensory data, without the need for human-crafted representations or prior expert demonstrations. Our method, which we dub **MERL**: **M**ultimodal **E**nd-to-end **R**einforcement **L**earning, results in a simple but effective approach capable of significantly outperforming both current state-of-the-art visual RL and state-based RL methods with respect to sample efficiency, learning performance, and training stability in relation to 3D robotic manipulation tasks from DeepMind Control.

## 1  Introduction

Deep reinforcement learning (deep RL), the effective combination of RL and deep learning, has allowed RL methods to attain remarkable results across a wide range of domains, including board and video games under discrete action space (Mnih et al., 2015; Silver et al., 2017; Vinyals et al., 2019) and robotics and control under continuous action space (Levine et al., 2016; Zhu et al., 2020; Kalashnikov et al., 2021; Ibarz et al., 2021; Kroemer et al., 2021). Many deep RL studies use human-crafted representations, for it is commonly known that state-based training operating on a coordinate state is significantly more sample-efficient than raw sensory data–based training (for example, image-based training). However, the use of human-crafted representations poses several major limitations and issues for 3D robotic manipulation: (a) human-crafted representations cannot perfectly represent the robot environment; (b) in the case of real-world applications, a separate module for environmental perception is required to obtain the environment state; and (c) state-based training based on human-crafted representations is not capable of using a neural network architecture repeatedly for different tasks, even for small variations in the environment such as changes in the number, size, or shape of objects.

Over the last three years, the RL community has made significant headway on these limitations and issues by significantly improving sample efficiency in image-based training (Hafner et al., 2019a;b; 2020; 2022; Lee et al., 2020a; Srinivas et al., 2020; Yarats et al., 2020; 2021a;b;c). A key insight of such studies is that the learning of better low-dimensional representations from image pixels is achieved through an autoencoder (Yarats et al., 2021c), variational inference (Hafner et al., 2019a;b; 2020; 2022; Lee et al., 2020a), contrastive learning (Srinivas et al., 2020; Yarats et al., 2021b), or

data augmentation (Yarats et al., 2020; 2021a), which in turn has helped to improve sample efficiency significantly. Recently, some visual RL studies have solved 3D continuous control problems such as quadruped and humanoid locomotion tasks from the DeepMind Control (DMC) suite, which in turn has helped to bridge the gap between state-based and image-based training (Hafner et al., 2020; Yarats et al., 2021a). Despite such significant progress in visual RL, image-based training is still notably sample-inefficient with respect to learning in 3D continuous control problems (for example, robotic manipulation) compared to state-based training.

In this paper, we propose an effective deep RL method for 3D robotic manipulation, which we dub **MERL**: **M**ultimodal **E**nd-to-end **R**einforcement **L**earning. MERL is a simple but effective model-free RL method that can be trained in an end-to-end manner from two different types of raw sensory data (RGB images and proprioception) having a multimodality that differs in dimensionality and value range. Most notably, MERL is capable of learning a latent multimodal representation and a policy in an efficient, joint, and end-to-end manner from multimodal raw sensory data, without the need for human-crafted representations or prior expert demonstrations. Compared to current state-of-the-art visual RL and state-based RL methods, MERL provides significant improvements in sample efficiency, learning performance, and training stability in relation to five 3D robotic manipulation tasks from DMC (*jaco-reach-duplo, jaco-move-box, jaco-lift-box, jaco-push-box-with-obstacle*, and *jaco-pick-and-stack*) (Tunyasuvunakool et al., 2020). In addition, MERL solves each of the three complex 3D humanoid locomotion tasks from DMC (*humanoid-stand*, *humanoid-walk*, and *humanoid-run*) within 5M environment steps, whereas the state-of-the-art visual RL, DrQ-v2 (Yarats et al., 2021a), solves the same tasks within 30M environment steps. To the best of our knowledge, MERL is the first model-free off-policy method not only to learn a latent multimodal representation and a policy in an efficient, joint, and end-to-end manner from multimodal raw sensory data, but also to show a new state-of-the-art performance by significantly outperforming both current state-of-the-art visual RL and state-based RL methods with respect to sample efficiency, learning performance, and training stability.

The main contributions of the paper can be summarized as follows: (1) the introduction of an end-to-end approach learning directly from multimodal raw sensory data for the efficient learning of a policy for use in the field of 3D robotic manipulation; (2) a demonstration of the fact that the approach significantly outperforms current state-of-the-art visual RL and state-based RL methods with respect to sample efficiency, learning performance, and training stability in relation to five 3D robotic manipulation tasks from DMC; (3) a demonstration of the superiority of the approach through the performance of complex 3D humanoid locomotion tasks from DMC within 5M environment steps; and (4) the provision of a deep RL method capable of learning a latent multimodal representation and a policy in an efficient, joint, and end-to-end manner from multimodal raw sensory data, without the need for human-crafted representations or prior expert demonstrations.

## 2 RELATED WORK

### 2.1 REINFORCEMENT LEARNING FOR 3D ROBOTIC MANIPULATION

Deep RL has seen widespread success across a variety of domains, including board and video games and robotics and control (Mnih et al., 2015; Silver et al., 2017; Vinyals et al., 2019; Levine et al., 2016; Zhu et al., 2020; Kalashnikov et al., 2021; Ibarz et al., 2021; Kroemer et al., 2021). In recent years, a number of deep RL methods have been successfully applied to 3D robotic manipulation tasks ranging from 'easy' (for example, *reach-target*) to 'hard' (for example, *assembly*). Of these methods, some require hand-engineered components for perception, state estimation, and low-level control, for they learn from human-crafted representations (for example, (Yamada et al., 2020; Lee et al., 2019b; 2021b; Nam et al., 2022)), and such are commonly referred to as state-based RL.

Over the last three years, visual RL that learns directly from image pixels has been greatly advanced and has seen significant improvements in sample efficiency in 3D continuous control problems, which in turn has helped to bridge the gap between state-based and image-based training (Hafner et al., 2019a;b; 2020; 2022; Srinivas et al., 2020; Yarats et al., 2020; 2021a;b;c; Wu et al., 2022). A key insight of such studies is that the learning of better low-dimensional representations from image pixels is achieved through various techniques such as an autoencoder, variational inference, contrastive learning, or data augmentation. Currently, DrQ-v2 (Yarats et al., 2021a) shows a state-of-the-art

performance in visual RL studies by solving complex 3D locomotion tasks from the DMC suite (Tassa et al., 2018) such as *humanoid locomotion*, previously unattained by model-free visual RL.

In this study, we propose MERL, a deep RL method for 3D robotic manipulation capable of learning a latent multimodal representation and a policy in an efficient, joint, and end-to-end manner from multimodal raw sensory data, including RGB images and proprioception, without the need for human-crafted representations or prior expert demonstrations. Here, the RGB images and proprioception are obtained from a fixed vision camera and a robot's joint encoders, respectively. Experimental results on 3D robotic manipulation tasks from DMC (Tunyasuvunakool et al., 2020) show that MERL significantly improves sample efficiency compared to state-based RL and DrQ-v2 (Yarats et al., 2021a), the current state-of-the-art visual RL.

## 2.2 REPRESENTATION LEARNING FOR MULTIMODAL INPUTS

The complementary nature of heterogeneous sensor modalities, such as vision, audio, language, haptic, range, and proprioceptive data, has previously been explored with respect to perception and decision-making. For instance, many studies have explored the correlation between such different modalities, including the correlation between visual and auditory data in relation to speech recognition (Yang et al., 2017; Afouras et al., 2018) or sound source localization (Tian et al., 2018; Owens & Efros, 2018); the correlation between visual and haptic data for grasping (Calandra et al., 2018; Narita & Kroemer, 2021) or manipulation (Lee et al., 2019a; 2020b; 2021a); the correlation between visual and tactile data for object tracking (Yu & Rodriguez, 2018; Lambert et al., 2019) or shape completion (Wang et al., 2018); and the correlation between visual and ranging data for robot navigation (Liu et al., 2017; Cai et al., 2020). While many of these studies have contributed to improvements in perception performance through multimodal representation learning, in this study, we are interested in improving the performance of RL through multimodal representation learning; in particular, the efficient joint learning of a latent multimodal representation and a policy.

In addition, some neuroscience studies have proved that the interdependence and concurrency of different sensory inputs aid perception and manipulation (Edelman, 1987; Lacey & Sathian, 2016; Bohg et al., 2017). Accordingly, we use two heterogeneous sensory data: RGB images and proprioception (for example, joint angles and velocities) to efficiently learn a policy for 3D robotic manipulation.

## 3 BACKGROUND

### 3.1 END-TO-END REINFORCEMENT LEARNING FROM MULTIMODAL RAW SENSORY DATA

We formulate continuous control problems on the basis of multimodal raw sensory data (that is, RGB image and proprioception) as an infinite-horizon Markov decision process (MDP) (Bellman, 1957). In such a setting, we stack three consecutive prior RGB images to properly approximate the environment's underlying state (Mnih et al., 2013). Accordingly, such an MDP can be described as a tuple $(\mathcal{O}, \mathcal{A}, P, R, \gamma, p_0)$, where $\mathcal{O} = \{\mathcal{O}^{\text{image}}, \mathcal{O}^{\text{prop}}\}$ denotes the state space, including the three consecutive high-dimensional RGB images, $\mathcal{O}^{\text{image}}$, and the low-dimensional proprioception vector, $\mathcal{O}^{\text{prop}}$; $\mathcal{A}$ denotes the action space; $P : \mathcal{O} \times \mathcal{A} \to \Delta(\mathcal{O})$ denotes the transition dynamics that define a probability distribution over the next state given the current state and action; $R : \mathcal{O} \times \mathcal{A} \to [0, 1]$ denotes the reward function that maps the current state and action to a reward; $\gamma \in [0, 1)$ denotes a discount factor; and $p_0 \in \Delta(\mathcal{O})$ denotes the probability distribution of the initial state $o_0$. The goal is to find a policy, $\pi : \mathcal{O} \to \Delta(\mathcal{A})$, that maximizes the expected discounted sum of rewards, $\mathbb{E}_\pi \left[ \sum_{t=1}^\infty \gamma^t r_t \right]$, where $o_0 \sim p_0$, and $\forall t$ we have $a_t \sim \pi(\cdot \mid o_t)$, $o_{t+1} \sim p(\cdot \mid o_t, a_t)$, and $r_t = R(o_t, a_t)$.

### 3.2 DEEP DETERMINISTIC POLICY GRADIENT

Deep Deterministic Policy Gradient (DDPG) (Lillicrap et al., 2015) is an off-policy actor–critic RL algorithm for continuous control that concurrently learns a Q-function, $Q_\phi$, and a deterministic policy, $\pi_\theta$. For this, DDPG uses Q-learning (Watkins & Dayan, 1992) to learn $Q_\phi$ by minimizing the one-step Bellman residual $J_\phi(\mathcal{D}) = \mathbb{E}_{(o_t, a_t, r_t, o_{t+1}) \sim \mathcal{D}} \left[ (Q_\phi(o_t, a_t) - r_t - \gamma Q_{\bar{\phi}}(o_{t+1}, \pi_\theta(o_{t+1})))^2 \right]$. The policy $\pi_\theta$ is learned by employing Deterministic Policy Gradient (DPG) (Silver et al., 2014) and

maximizing $J_\theta(\mathcal{D}) = \mathbb{E}_{o_t \sim \mathcal{D}} [Q_\phi(o_t, \pi_\theta(o_t))]$, where $\pi_\theta(o_t)$ approximates $\texttt{argmax}_a Q_\phi(o_t, a)$. Here, $\mathcal{D}$ is a replay buffer of environment transitions and $\bar\phi$ is an exponential moving average of the weights. DDPG is amenable to incorporate $n$-step returns when estimating temporal difference (TD) error beyond a single step. In practice, $n$-step returns allow for faster reward propagation and have been previously used in policy gradient and Q-learning methods (Hessel et al., 2018; Barth-Maron et al., 2018; Li & Faisal, 2021).

## 3.3 SOFT ACTOR-CRITIC

Soft actor-critic (SAC) (Haarnoja et al., 2018a;b) is an off-policy actor–critic RL algorithm for continuous control problems that concurrently learns a Q-function, $Q_\phi$, a stochastic policy, $\pi_\theta$, and a temperature, $\alpha$, on the basis of a maximum-entropy framework. For this, SAC performs soft policy evaluation and improvement steps at each iteration, with the goal of maximizing a trade-off between expected return and entropy using the following $\gamma$-discounted maximum-entropy objective: $\mathbb{E}_{(o_t, a_t) \sim \mathcal{D}} [R(o_t, a_t) + \alpha H(\pi(\cdot \mid o_t))]$, where $\alpha$ is the temperature that balances between optimizing for the reward and for the stochasticity of the policy.

## 3.4 IMAGE AUGMENTATION IN REINFORCEMENT LEARNING

Image augmentation techniques have been commonly used in computer vision research not only to avoid overfitting, but also to achieve state-of-the-art performance (Shorten & Khoshgoftaar, 2019). Recently, in visual RL, DrQ-v2 (Yarats et al., 2021a), which is an improvement on DrQ (Yarats et al., 2020), provides a new state-of-the-art performance in relation to 3D locomotion tasks from the DMC suite (Tassa et al., 2018) by adding image augmentation in the form of random shifts. The use of such image augmentation techniques is now an essential factor in achieving a new state-of-the-art performance in image-based training (Srinivas et al., 2020; Schwarzer et al., 2020; Yarats et al., 2020; 2021a; Hansen & Wang, 2021).

# 4 MERL: MULTIMODAL END-TO-END REINFORCEMENT LEARNING

In this section, we describe MERL, an effective deep RL method for 3D robotic manipulation; specifically, a model-free off-policy actor–critic RL algorithm that can be trained in an end-to-end manner from multimodal raw sensory data (high-dimensional RGB images and a low-dimensional proprioception vector). Figure 1 illustrates our method (MERL), which learns a latent multimodal representation and a policy in an efficient, joint, and end-to-end manner from multimodal raw sensory data, without the need for human-crafted representations or prior expert demonstrations. Here, MERL uses an image augmentation technique and a convolutional neural network (CNN) to encode the images; a multi-layer perception (MLP) to encode the proprioception; an MLP and layer normalization with scaling under a decoupled architecture to learn the latent multimodal representation from the encoded visual and proprioceptive representations; and actor–critic networks with layer normalization applied to learn the policy.

## 4.1 MULTIMODAL REPRESENTATION LEARNING FOR 3D ROBOTIC MANIPULATION

**Replay Buffer** Unlike in visual RL such as DrQ-v2 or state-based RL such as DDPG and SAC, our method stores transition $(\boldsymbol{o}_t, \boldsymbol{a}_t, r_t, \gamma_t, \boldsymbol{o}_{t+1})$, including multimodal observations, in the replay buffer, for we use two different types of raw sensory data (RGB images and proprioception). Regarding the transition, $\boldsymbol{o}_t$ denotes a multimodal observation (three consecutive prior RGB images, $\boldsymbol{o}_t^{\text{image}}$, and a proprioception vector, $\boldsymbol{o}_t^{\text{prop}}$) at time $t$; $\boldsymbol{a}_t$ denotes the robot's action (for example, joint torques) at time $t$; $r_t$ denotes the reward at time $t$; $\gamma_t$ denotes a discount factor at time $t$; and $\boldsymbol{o}_{t+1}$ denotes a multimodal observation (three consecutive prior RGB images, $\boldsymbol{o}_{t+1}^{\text{image}}$, and a proprioception vector, $\boldsymbol{o}_{t+1}^{\text{prop}}$) at time $t + 1$. Regarding the proprioception vector, it can include the joint positions, velocities, or torques of the robotic arm and hand or the position or rotation of the robot's end-effector, or all of the above.

**Image Augmentation** A deep neural network is a powerful tool with which to learn a latent representation from high-dimensional data; however, such a tool requires a substantial amount of

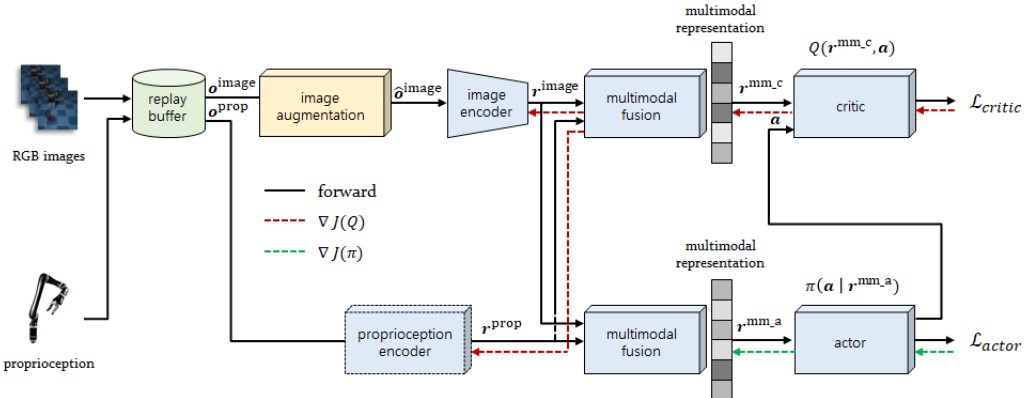

Figure 1: Our method (MERL) learns a latent multimodal representation and a policy in an efficient, joint, and end-to-end manner from multimodal raw sensory data (high-dimensional RGB images and a low-dimensional proprioception vector), without the need for human-crafted representations or prior expert demonstrations. A new state-of-the-art performance comes from a careful configuration of multimodal representation learning combined with data-augmented RL.

training data. The computer vision community has made great headway regarding this challenge by using various image augmentation techniques (Shorten & Khoshgoftaar, 2019). In this study, we apply a random shift image augmentation technique to image observations of the environment, as in DrQ-v2. Specifically, we pad each side of an $84\times84$ image with four repeating boundary pixels. Next, we replace each repeated pixel value with the average of the four nearest pixel values (that is, we apply bilinear interpolation to the shifted image). Finally, we select a random $84\times84$ crop, yielding the original image shifted by $\pm4$ pixels.

**Image Encoder**    The augmented images, $\hat{o}^{\text{image}}$, obtained by means of the random shift augmentation technique, are embedded into a low-dimensional latent vector, $r^{\text{image}}$, by means of a CNN encoder. Regarding the CNN encoder, we use the same architecture as in DrQ (Yarats et al., 2020), which was first introduced in SAC+AE (Yarats et al., 2021c). This is due to the fact that DrQ has successfully shown a range of encoders of differing architecture and capacity to perform equally well on continuous control problems in the case where a random shift augmentation technique is applied to original image observations. The encoding process can be concisely formulated as $r^{\text{image}} = f_\xi(\texttt{aug}(o^{\text{image}}))$, where $f_\xi$ denotes the image encoder, $\texttt{aug}$ denotes the random shift augmentation, and $o^{\text{image}}$ denotes the original image observation.

**Proprioception Encoder**    Regarding the proprioception encoder, we use a simple MLP. The encoding process can be concisely summarized as $r^{\text{prop}} = g_\zeta(o^{\text{prop}})$, where $g_\zeta$ denotes the proprioception encoder and $o^{\text{prop}}$ denotes the original proprioception observation. Note here that the best performance is achieved when the proprioception encoder is set as the $\texttt{identity}$ encoder. This is due to the fact that the proprioception is already well-encoded by itself (see Appendix C.1 for more details).

**Multimodal Fusion**    MERL uses two different types of raw sensory data as input: the RGB images from a fixed vision camera and the proprioception from the joint encoders of the robot arm and hand. The heterogeneous nature of this data requires us to use domain-specific encoders to obtain the unique characteristics of each modality, which we then fuse into a single latent multimodal representation vector, $r^{\text{mm}}$, of dimension $\texttt{d}$.

For visual feedback, we stack three consecutive prior RGB images, apply the random shift augmentation technique to the stacked images, normalize the resulting augmented images, and use a CNN to encode the normalized images into a latent representation vector, $r^{\text{image}}$. For proprioception, we use an MLP to encode the proprioception into a latent representation vector, $r^{\text{prop}}$. For multimodal fusion, we use an MLP and layer normalization with scaling to produce a low-dimensional scaled latent vector, of which the range is $(-1, 1)$ for each modality. Each scaled latent vector is concatenated to produce a single latent multimodal representation vector, $r^{\text{mm}}$ of dimension $\texttt{d}$. The multimodal fusion

process can be succinctly summarized as $\boldsymbol{r}^{\text{mm}} = h_\psi(\boldsymbol{r}^{\text{image}}, \boldsymbol{r}^{\text{prop}})$, where $\boldsymbol{r}^{\text{image}} = f_\xi(\text{aug}(\boldsymbol{o}^{\text{image}}))$ denotes the encoded visual representation from the RGB images and $\boldsymbol{r}^{\text{prop}} = g_\zeta(\boldsymbol{o}^{\text{prop}})$ denotes the encoded proprioceptive representation from the proprioception. Note here that we apply a decoupled architecture to multimodal fusion to improve the performance of actor–critic learning; that is, the decoupled architecture uses two different latent multimodal representation vectors, $\boldsymbol{r}^{\text{mm\_a}}$ and $\boldsymbol{r}^{\text{mm\_c}}$, to train the actor and critic networks separately. As a result, each of the representations are learned to better optimize the losses of the actor and critic, respectively.

### 4.2 MODEL-FREE ACTOR–CRITIC REINFORCEMENT LEARNING

Although SAC has been the de facto off-policy RL algorithm for many RL methods over the past few years, it is prone to suffering from policy entropy collapse. Recently, DrQ-v2 showed that using DDPG instead of SAC as a learning algorithm leads to better performance (that is, more robustness and more stability) in relation to the tasks from the DMC suite, including 3D continuous control problems. For this reason, we opt for DDPG as a backbone actor–critic RL algorithm to learn a policy from a latent multimodal representation. As in DrQ-v2, we also augment DDPG with $n$-step returns to estimate TD error: the augmentation leads to faster learning progress by accelerating reward propagation. Here, we opt for augmenting DDPG with 3-step returns. For the augmentation, we sample a mini-batch of transitions, $\tau = (\boldsymbol{o}_t, \boldsymbol{a}_t, r_{t:t+n-1}, \boldsymbol{o}_{t+n})$, from the replay buffer, $\mathcal{D}$. In addition, we use clipped double Q-learning to reduce over-estimation bias in the target value. For the learning, we train two Q-functions, $Q_{\phi_1}$ and $Q_{\phi_2}$, by minimizing the following two losses:

$$\mathcal{L}_Q(\phi_k, \psi_{\text{critic}}, \xi, \zeta, \mathcal{D}) = \mathbb{E}_{(\boldsymbol{o}_t, \boldsymbol{a}_t, r_{t:t+n-1}, \boldsymbol{o}_{t+n}) \sim \mathcal{D}} \left[ (Q_{\phi_k}(\boldsymbol{r}_t^{\text{mm\_c}}, \boldsymbol{a}_t) - y)^2 \right] \quad \forall k \in \{1, 2\}, \quad (1)$$

where TD target, $y$, is defined as follows:

$$y = \sum_{i=0}^{n-1} \gamma^i r_{t+i} + \gamma^n \min_{k=1,2} Q_{\bar{\phi}_k}(\boldsymbol{r}_{t+n}^{\text{mm\_c}}, \boldsymbol{a}_{t+n}),$$

where $\boldsymbol{r}_t^{\text{mm\_c}} = h_{\psi_{\text{critic}}}(f_\xi(\text{aug}(\boldsymbol{o}_t^{\text{image}})), g_\zeta(\boldsymbol{o}_t^{\text{prop}}))$, $\boldsymbol{r}_{t+n}^{\text{mm\_c}} = h_{\psi_{\text{critic}}}(f_\xi(\text{aug}(\boldsymbol{o}_{t+n}^{\text{image}})), g_\zeta(\boldsymbol{o}_{t+n}^{\text{prop}}))$, $\boldsymbol{a}_t = \pi_\theta(\boldsymbol{r}_t^{\text{mm\_a}}) + \epsilon$, and $\boldsymbol{a}_{t+n} = \pi_\theta(\boldsymbol{r}_{t+n}^{\text{mm\_a}}) + \epsilon$. Here, $\epsilon$ represents exploration noise sampled from clip($\mathcal{N}(0, \sigma^2), -c, c$), which is similar to TD3 (Fujimoto et al., 2018); $\bar{\phi}_1$ and $\bar{\phi}_2$ represent the exponential moving averages of the weights for the Q target networks; and $\psi_{\text{critic}}$ represents the multimodal fusion weights for the critic. The final version of MERL uses decoupled multimodal fusion networks for the actor and critic, $h_{\psi_{\text{actor}}}$ and $h_{\psi_{\text{critic}}}$, respectively. We note here that we use the most recent weights of $\xi$ for the image encoder, $\zeta$ for the proprioception encoder, and $\psi$ for the multimodal fusion, to encode $\boldsymbol{o}_t$ and $\boldsymbol{o}_{t+n}$. Finally, we train the deterministic actor, $\pi_\theta$, using DPG by maximizing the expected returns, as follows:

$$\mathcal{L}_\pi(\theta, \psi_{\text{actor}}, \mathcal{D}) = -\mathbb{E}_{\boldsymbol{o}_t \sim \mathcal{D}} \left[ \min_{k=1,2} Q_{\phi_k}(\boldsymbol{r}_t^{\text{mm\_c}}, \boldsymbol{a}_t) \right], \quad (2)$$

where $\boldsymbol{r}_t^{\text{mm\_a}} = h_{\psi_{\text{actor}}}(f_\xi(\text{aug}(\boldsymbol{o}_t^{\text{image}})), g_\zeta(\boldsymbol{o}_t^{\text{prop}}))$ and $\boldsymbol{a}_t = \pi_\theta(\boldsymbol{r}_t^{\text{mm\_a}}) + \epsilon$. Here, $\psi_{\text{actor}}$ represents the multimodal fusion weights for the actor. We note here that we prevent the actor's gradients from updating the image and proprioception encoders, for SAC+AE found that this in fact hinders the agent's performance.

## 5 EXPERIMENTS

In this section, we provide an empirical evaluation of MERL in relation to five 3D robotic manipulation tasks from DMC (*jaco-reach-duplo, jaco-move-box, jaco-lift-box, jaco-push-box-with-obstacle*, and *jaco-pick-and-stack*) (Tunyasuvunakool et al., 2020). We first present a comparison to prior methods, including current state-of-the-art visual RL and state-based RL methods, with respect to sample efficiency, learning performance, and training stability. We then present an ablation study that guided the final version of MERL.

### 5.1 SETUP

**Environment Setup** We consider an environment for 3D robotic manipulation that provides multimodal raw sensory data as observation. In this context, we consider learning directly from

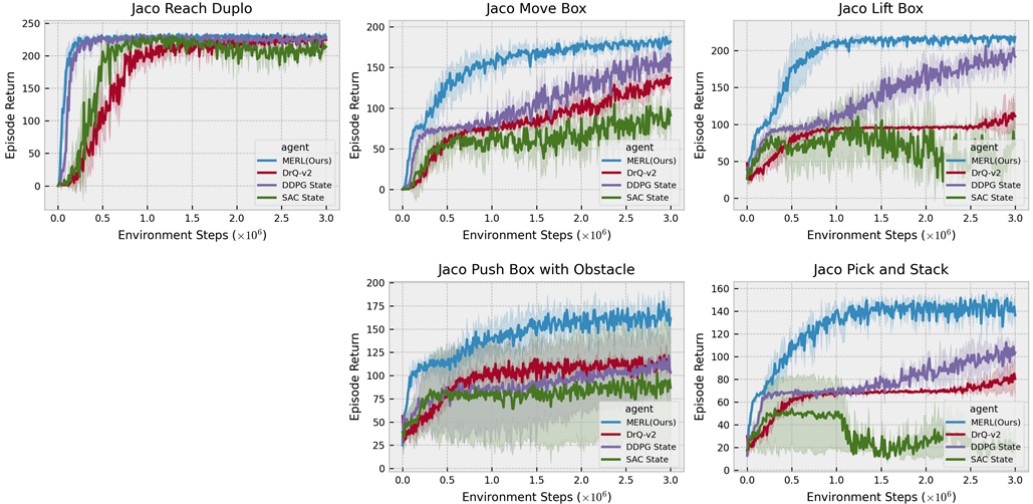

Figure 2: We compare MERL to both current state-of-the-art visual RL and state-based RL methods in relation to *five* 3D robotic manipulation tasks from DMC (*jaco-reach-duplo, jaco-move-box, jaco-lift-box, jaco-push-box-with-obstacle,* and *jaco-pick-and-stack*). MERL demonstrates superior sample efficiency, significantly outperforms leading model-free RL baselines, and shows more stable training performance.

multimodal observations, including RGB images and proprioception. Here, the RGB images represent stacks of three consecutive RGB images of size $84 \times 84$, stacked along the channel dimension to enable inference of dynamic information such as velocity and acceleration, and the proprioception represents joint angles and velocities for the robotic arm and hand. The action space is bounded by $(-1, 1)$.

**Task Setup** We design a set of *five* 3D robotic manipulation tasks from DMC (*jaco-reach-duplo, jaco-move-box, jaco-lift-box, jaco-push-box-with-obstacle,* and *jaco-pick-and-stack*) to include two different types of raw sensory data (that is, RGB images obtained from a fixed vision camera and proprioception obtained from a robot's joint encoders) as observation, where task success requires joint reasoning over visual and proprioceptive feedback. In addition, for each task, we randomize the configuration of the initial positions of the robot and the box at the beginning of each episode, during both training and testing, to enhance the robustness and generalization of the model.

**Reward Design** We adopt a staged, structured, and multi-component reward function to guide the RL algorithm, which simplifies the challenge of exploration and leads to effective policy learning (Lee et al., 2020b; Yu et al., 2020). The reward function, R, is a combination of a reaching reward, pushing reward, vertical reward, floating reward, and lifting reward, or subsets thereof for simpler tasks that only include reaching or pushing (see Appendix B.2 for more details). With this design, the reward is bounded by [0, 1] per timestep.

## 5.2 COMPARISON TO PRIOR METHODS

**Baselines** We compare our method to DrQ-v2 (Yarats et al., 2021a), which currently provides the best performance for 3D continuous control problems within model-free visual RL methods, with respect to sample efficiency, learning performance, and training stability. We also compare it to the state-based RL algorithms DDPG (Fujimoto et al., 2018) and SAC (Haarnoja et al., 2018b), which provide upper-bound performance with respect to sample efficiency for many RL studies.

**Evaluation** We present our experimental results for *five* 3D robotic manipulation tasks from DMC (*jaco-reach-duplo, jaco-move-box, jaco-lift-box, jaco-push-box-with-obstacle,* and *jaco-pick-and-stack*), in Figure 2. Our empirical study in Figure 2 reveals that MERL considerably outperforms both the state-based RL algorithms (DDPG and SAC) and the state-of-the-art visual RL algorithm

(DrQ-v2) with respect to sample efficiency, learning performance, and training stability. This suggests that a coordinate state used in state-based RL is insufficient to represent the robot environment and that visual RL requires significantly more data to learn a latent representation, both of which in turn suggest that learning from multimodal raw sensory data is required to achieve the best performance with respect to sample efficiency and learning performance in relation to 3D robotic manipulation. Notably, MERL's advantage is more pronounced in the case of more difficult 3D robotic manipulation tasks (for example, *jaco-move-box, jaco-lift-box, jaco-push-box-with-obstacle*, and *jaco-pick-and-stack*), where exploration is more challenging. Most notably, to the best of our knowledge, MERL is the first model-free RL method for 3D robotic manipulation capable of learning a latent multimodal representation and a policy in an efficient, joint, and end-to-end manner from multimodal raw sensory data, without the need for human-crafted representations or prior expert demonstrations, while showing a new state-of-the-art performance.

## 5.3  ABLATION STUDY

We conduct an ablation study that leads to the final version of MERL. Our findings are summarized in Figure 3 and detailed below.

**Decoupled Multimodal Fusion**   We examine our decision to use a decoupled architecture for multimodal fusion compared to using a conventional shared architecture primarily used in multimodal representation learning (Lee et al., 2019a; 2020b). We hypothesize that the performance of actor–critic learning could be improved by feeding multiple representations into the actor and critic separately and learning these representations in such a manner so as to better optimize the losses of the actor and critic, respectively. We conduct some experiments on two different architectures for multimodal fusion: decoupled and shared. As shown in Figure 3a, the decoupled architecture provides better performance in relation to the three 3D robotic manipulation tasks from DMC (*jaco-reach-duplo, jaco-move-box*, and *jaco-lift-box*) compared to the conventional shared architecture. This suggests that the decoupled architecture for multimodal fusion has a greater impact on actor–critic learning compared to the conventional shared architecture. The greater impact may be due to the fact that in the decoupled architecture latent multimodal representations fed into the actor are separately learned to maximize the learning performance of the actor, and likewise in the case of the critic. To the best of our knowledge, MERL is the first RL method to apply a decoupled architecture to actor–critic learning. We also found that, in the case of the shared architecture for multimodal fusion, updating the multimodal fusion with the actor's gradients greatly hinders the agent's performance.

**Layer Normalization**   We examine the efficacy of layer normalization (Ba et al., 2016) commonly used in transformer architectures (Vaswani et al., 2017; Wang et al., 2019). We hypothesize that the application of layer normalization to our multimodal fusion will play a crucial role in controlling the gradient scales, as in a transformer architecture, and will lead to favorable sample efficiency. We conduct some experiments for two different cases: multimodal fusion with and without layer normalization applied. As shown in Figure 3b, the former case produces significantly better performance in relation to the three 3D robotic manipulation tasks from DMC (*jaco-reach-duplo, jaco-move-box*, and *jaco-lift-box*) than the latter case. This suggests that any increase in speed or stability of the RL process is due to the use of normalized multimodal representations (via layer normalization) in actor–critic networks. In light of this, we further proceed to study whether the use of layer normalization in other baselines would have a similar effect on performance; indeed, we found that the use of layer normalization in RL algorithms is crucial to the study of RL (see Appendix E). Note that the final version of MERL uses layer normalization in both multimodal fusion and actor–critic networks (see Appendix C.1). In addition, we perform scaling via `tanh` before feeding the representation into the critic network in order to match the latent multimodal representation and action to the same scale. We note that the use of layer normalization in RL algorithms has received little attention in the RL community.

**Multimodal Representation Dimensions**   We investigate the required level of compactness of the latent multimodal representation for 3D robotic manipulation by changing the dimensionality of the representation. We hypothesize that a more compact representation may increase the tractability of RL yet capture less information for 3D robotic manipulation tasks. We perform several experiments at different representation dimensions: `dim` = 64, 128, and 256. As shown in Figure 3c, `dim` = 128

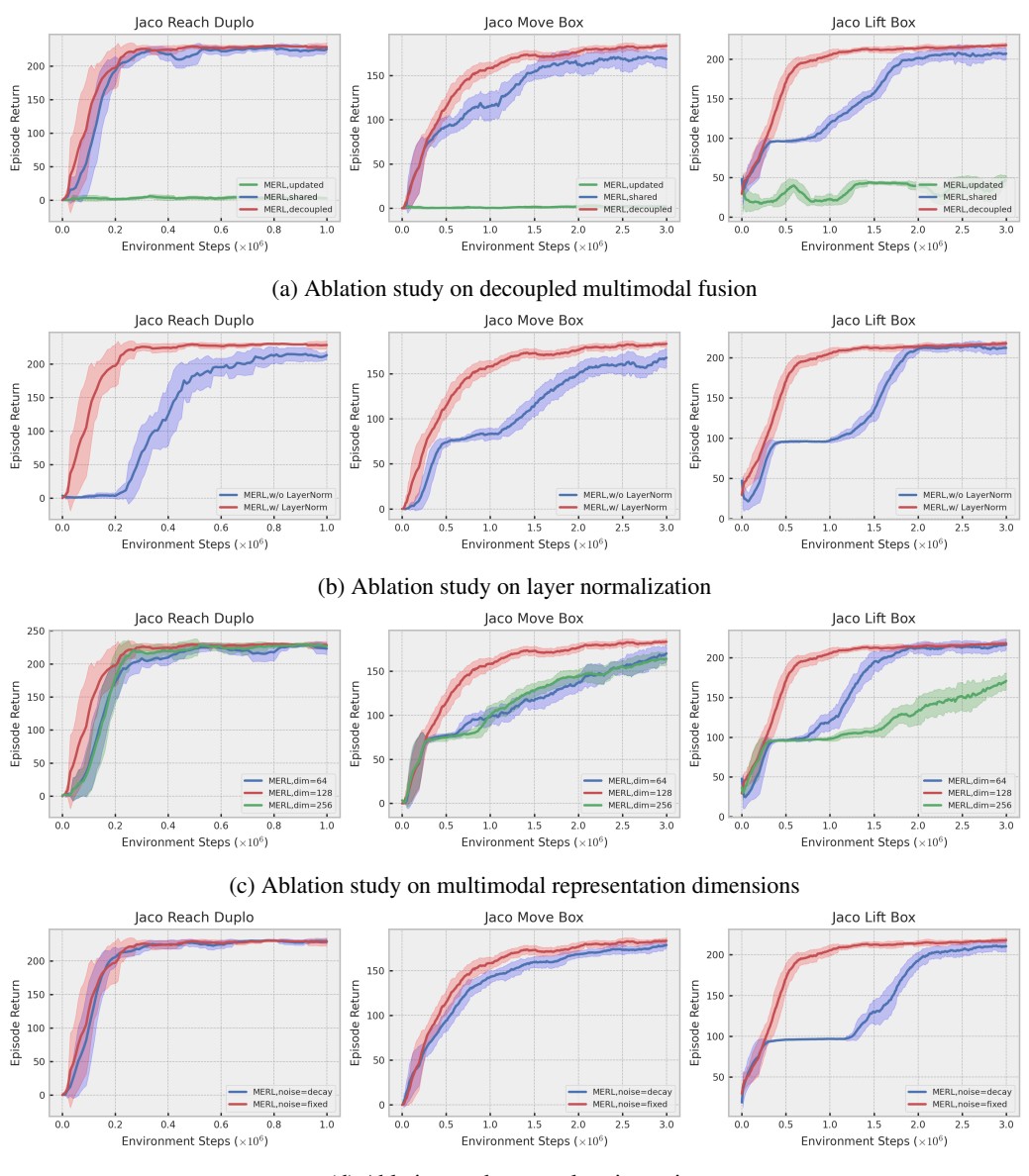

Figure 3: An ablation study that led us to the final version of MERL: (a) we observe that the decoupled architecture for multimodal fusion (*red*) provides better performance compared to the conventional shared architecture (*blue*); (b) we observe that the use of layer normalization in multimodal fusion (*red*) results in a significant performance gain in sample efficiency; (c) we observe that `dim` = 128 for multimodal representation (*red*) provides the best performance for the given 3D robotic manipulation tasks; and (d) we observe that the fixed stochastic exploration (*red*) provides better performance for 3D robotic manipulation compared to the scheduled exploration (*blue*) used in DrQ-v2.

provides the best performance in relation to the three 3D robotic manipulation tasks from DMC (*jaco-reach-duplo, jaco-move-box*, and *jaco-lift-box*). In the case of `dim` = 64, less information for the tasks was captured, and in the case of `dim` = 256, the learning performance was degraded owing to an increase in the latent state space used in policy learning.

**Exploration Noise** We investigate the efficacy of the scheduled exploration noise used in DrQ-v2, where the scheduled exploration noise has different levels of exploration at different stages of learning. We hypothesize that a scheduled exploration noise will be more helpful to improve sample efficiency

by having a more stochastic exploration at the beginning of training compared to a fixed exploration noise. We perform some experiments at two different levels of exploration noise: decaying and fixed. As shown in Figure 3d, contrary to the results of the DrQ-v2 study (Yarats et al., 2021a), having a fixed level of exploration noise provides better learning performance in relation to the three 3D robotic manipulation tasks from DMC (*jaco-reach-duplo, jaco-move-box*, and *jaco-lift-box*) than having a scheduled level of exploration noise. This suggests that the data accumulated through having a stochastic exploration at the beginning of training degrades the learning performance, especially in the case of the task *jaco-lift-box*. Accordingly, we need to set different levels of exploration noise depending on the characteristics of a given task. For example, for a task that requires a lot of stochastic exploration at the beginning of training, such as *humanoid locomotion*, a scheduled exploration noise is more effective, whereas, in the case of 3D robotic manipulation tasks, a fixed stochastic exploration is more effective.

# 6  CONCLUSION

In this study, we have proposed MERL, a conceptually simple but effective model-free off-policy RL method for 3D robotic manipulation capable of learning a latent multimodal representation and a policy in an efficient, joint, and end-to-end manner from multimodal raw sensory data (RGB images and proprioception), without the need for human-crafted representations or prior expert demonstrations. Our experimental results show that MERL significantly outperforms both current state-of-the-art visual RL and state-based RL methods with respect to sample efficiency, learning performance, and training stability in relation to 3D continuous control problems, including robotic manipulation and locomotion tasks from DMC. Most notably, MERL is able to solve the primitive skills *reach*, *push*, *move*, *lift*, and *pick-and-place* with respect to the chosen 3D robotic manipulation tasks from DMC, within 1M environment steps. Also, MERL is able to solve each of the three complex 3D humanoid locomotion tasks from DMC (*humanoid-stand*, *humanoid-walk*, and *humanoid-run*) — these tasks being among the most difficult of 3D continuous control problems — within 5M environment steps and without the need for human-crafted representations or prior expert demonstrations. Such experimental results demonstrate that MERL is one of the most effective methods for solving 3D continuous control problems, including robotic manipulation and locomotion tasks, with respect to learning a policy in an end-to-end manner, without the need for human-crafted representations or prior expert demonstrations. In addition, to the best of our knowledge, MERL is the first effective RL method for 3D robotic manipulation capable of learning a latent multimodal representation and a policy in an efficient, joint, and end-to-end manner from multimodal raw sensory data, without the need for human-crafted representations or prior expert demonstrations, while showing a new state-of-the-art performance. Furthermore, to the best of our knowledge, MERL is the first RL method to apply a decoupled architecture to actor–critic learning. We hope this work will serve as a guideline for future multimodal end-to-end RL research.

In future work, we will seek to use more sensors (for example, depth and haptic sensors) of differing modalities, which will lead to more dexterous manipulation, and to test and validate MERL in a real-world robotic environment through sim-to-real. In addition, we hope to explore the use of transformer architectures and object-centric learning to obtain a better representation.

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

# A ALGORITHM DETAILS

---

**Algorithm 1** MERL: Multimodal end-to-end deep RL

---

**Parametric networks:** Image encoder $f_\xi$, proprioception encoder $g_\zeta$, multimodal fusion for critic $h_{\psi_{\text{critic}}}$, multimodal fusion for actor $h_{\psi_{\text{actor}}}$, actor $\pi_\theta$, critic $Q_\phi$

**Hyper-parameters:** Training steps $T$, mini-batch size $B$, learning rate $\alpha$, target update rate $\tau$, standard deviation $\sigma$, clip value $c$

**Image augmentations:** Random shift aug

**for** $t \leftarrow 1 \cdots T$ **do**

    $\boldsymbol{a}_t \leftarrow \pi_\theta(h_{\psi_{\text{actor}}}(f_\xi(\boldsymbol{o}_t^{\text{image}}), g_\zeta(\boldsymbol{o}_t^{\text{prop}}))) + \epsilon$ and $\epsilon \sim \mathcal{N}(0, \sigma^2)$

    $\boldsymbol{o}_{t+1} \sim P(\cdot \mid \boldsymbol{o}_t, \boldsymbol{a}_t)$

    $\mathcal{D} \leftarrow (\boldsymbol{o}_t, \boldsymbol{a}_t, R(\boldsymbol{o}_t, \boldsymbol{a}_t), \boldsymbol{o}_{t+1}) \cup \mathcal{D}$

    UPDATECRITIC($\mathcal{D}$)

    UPDATEACTOR($\mathcal{D}$)

**end for**

**procedure** UPDATECRITIC($\mathcal{D}$)

    $\{(\boldsymbol{o}_t, \boldsymbol{a}_t, r_{t:t+n-1}, \boldsymbol{o}_{t+n})\} \sim \mathcal{D}$

    $\boldsymbol{r}_t^{\text{mm\_c}}, \boldsymbol{r}_{t+n}^{\text{mm\_c}} \leftarrow h_{\psi_{\text{critic}}}(f_\xi(\text{aug}(\boldsymbol{o}_t^{\text{image}})), g_\zeta(\boldsymbol{o}_t^{\text{prop}})), h_{\psi_{\text{critic}}}(f_\xi(\text{aug}(\boldsymbol{o}_{t+n}^{\text{image}})), g_\zeta(\boldsymbol{o}_{t+n}^{\text{prop}}))$

    $\boldsymbol{r}_t^{\text{mm\_a}}, \boldsymbol{r}_{t+n}^{\text{mm\_a}} \leftarrow h_{\psi_{\text{actor}}}(f_\xi(\text{aug}(\boldsymbol{o}_t^{\text{image}})), g_\zeta(\boldsymbol{o}_t^{\text{prop}})), h_{\psi_{\text{actor}}}(f_\xi(\text{aug}(\boldsymbol{o}_{t+n}^{\text{image}})), g_\zeta(\boldsymbol{o}_{t+n}^{\text{prop}}))$

    $\boldsymbol{a}_{t+n} \leftarrow \pi_\theta(\boldsymbol{r}_{t+n}^{\text{mm\_a}}) + \epsilon$ and $\epsilon \sim \text{clip}(\mathcal{N}(0, \sigma^2), -c, c)$

    Compute $\mathcal{L}_{\phi_1, \psi_{\text{critic}}, \xi, \zeta}$ and $\mathcal{L}_{\phi_2, \psi_{\text{critic}}, \xi, \zeta}$

    $\xi \leftarrow \xi - \alpha \nabla_\xi (\mathcal{L}_{\phi_1, \psi_{\text{critic}}, \xi, \zeta} + \mathcal{L}_{\phi_2, \psi_{\text{critic}}, \xi, \zeta})$

    $\zeta \leftarrow \zeta - \alpha \nabla_\zeta (\mathcal{L}_{\phi_1, \psi_{\text{critic}}, \xi, \zeta} + \mathcal{L}_{\phi_2, \psi_{\text{critic}}, \xi, \zeta})$

    $\psi_{\text{critic}} \leftarrow \psi_{\text{critic}} - \alpha \nabla_{\psi_{\text{critic}}}(\mathcal{L}_{\phi_1, \psi_{\text{critic}}, \xi, \zeta} + \mathcal{L}_{\phi_2, \psi_{\text{critic}}, \xi, \zeta})$

    $\phi_k \leftarrow \phi_k - \alpha \nabla_{\phi_k} \mathcal{L}_{\phi_k, \psi_{\text{critic}}, \xi, \zeta} \quad \forall k \in \{1, 2\}$

    $\bar{\phi}_k \leftarrow (1 - \tau)\bar{\phi}_k + \tau\phi_k \quad \forall k \in \{1, 2\}$

**end procedure**

**procedure** UPDATEACTOR($\mathcal{D}$)

    $\{(\boldsymbol{o}_t)\} \sim \mathcal{D}$

    $\boldsymbol{r}_t^{\text{mm\_c}} \leftarrow h_{\psi_{\text{critic}}}(f_\xi(\text{aug}(\boldsymbol{o}_t^{\text{image}})), g_\zeta(\text{o}_t^{\text{prop}}))$

    $\boldsymbol{r}_t^{\text{mm\_a}} \leftarrow h_{\psi_{\text{actor}}}(f_\xi(\text{aug}(\boldsymbol{o}_t^{\text{image}})), g_\zeta(\text{o}_t^{\text{prop}}))$

    $\boldsymbol{a}_t \leftarrow \pi_\theta(\boldsymbol{r}_t^{\text{mm\_a}}) + \epsilon$ and $\epsilon \sim \text{clip}(\mathcal{N}(0, \sigma^2), -c, c)$

    Compute $\mathcal{L}_{\theta, \psi_{\text{actor}}}$

    $\psi_{\text{actor}} \leftarrow \psi_{\text{actor}} - \alpha \nabla_{\psi_{\text{actor}}} \mathcal{L}_{\theta, \psi_{\text{actor}}}$

    $\theta \leftarrow \theta - \alpha \nabla_\theta \mathcal{L}_{\theta, \psi_{\text{actor}}}$

**end procedure**

---

## B    TASK DETAILS

In this section, we describe details of the five 3D robotic manipulation tasks from DMC (*jaco-reach-duplo, jaco-move-box*, *jaco-lift-box*, *jaco-push-box-with-obstacle*, and *jaco-pick-and-stack*) with respect to task descriptions, reward design, and task visualizations.

### B.1    TASK DESCRIPTIONS

We design a set of five 3D robotic manipulation tasks from DMC (*jaco-reach-duplo, jaco-move-box*, *jaco-lift-box*, *jaco-push-box-with-obstacle*, and *jaco-pick-and-stack*) (Tunyasuvunakool et al., 2020) to include two different types of raw sensory data (that is, RGB images obtained from a fixed vision camera and proprioception obtained from a robot's joint encoders) as observation, where task success requires joint reasoning over visual and proprioceptive feedback. For each task, we randomize the configuration of the initial positions of the robot and the box at the beginning of each episode, during both training and testing, to enhance the robustness and generalization of the model.

In the case of the task *jaco-reach-duplo*, the robot (more precisely, the center point of the robot's end-effector) is required to reach a duplo randomly placed on a workspace. The task is considered successful when the robot reaches within 5 cm of the duplo.

In the case of the task *jaco-move-box*, the robot is required to move a box randomly placed on a workspace to a specific target position. The task is considered successful when the box reaches within 1 cm of the target position.

In the case of the task *jaco-lift-box*, the robot is required to lift a box randomly placed on a workspace to a specific target position. The task is considered successful when the box reaches within 1 cm of the target position.

In the case of the task *jaco-push-box-with-obstacle*, the robot is required to push a box randomly placed on a workspace to a specific target position obscured by an obstacle. The task is considered successful when the box reaches within 1 cm of the target position.

In the case of the task *jaco-pick-and-stack*, the robot is required to lift a box randomly placed on a workspace and stack it on top of a cylinder. The task is considered successful when the box reaches within 1 cm of the cylinder's top center position.

### B.2    REWARD DESIGN

We adopt a staged, structured, and multi-component reward function to guide the RL algorithm, which simplifies the challenge of exploration and leads to effective policy learning (Lee et al., 2020b; Yu et al., 2020). The reward function, $R$, is a combination of a reaching reward, pushing reward, vertical reward, floating reward, and lifting reward, or subsets thereof for simpler tasks that only include reaching or pushing. With this design, the reward is bounded by $[0, 1]$ per timestep.

**Jaco Reach Duplo**    The reward function for task *jaco-reach-duplo* is defined as follows:

$$r_t = r_{\text{reaching}} \quad \text{(reaching)} \tag{3}$$

**Jaco Move Box**    The reward function for task *jaco-move-box* is defined as follows:

$$r_t = \begin{cases} r_{\text{reaching}} & \text{(reaching)} \\ r_{\text{reaching}} + r_{\text{pushing}} & \text{if } \text{dist}(\boldsymbol{p}_{\text{tcp}}, \boldsymbol{p}_{\text{obj}}) < \varepsilon_{\text{reaching}} \quad \text{(pushing)} \end{cases} \tag{4}$$

**Jaco Lift Box**    The reward function for task *jaco-lift-box* is defined as follows:

$$r_t = \begin{cases} r_{\text{reaching}} + r_{\text{vertical}} & \text{(reaching)} \\ r_{\text{reaching}} + r_{\text{vertical}} + r_{\text{floating}} & \text{if } \text{dist}(\boldsymbol{p}_{\text{tcp}}, \boldsymbol{p}_{\text{obj}}) < \varepsilon_{\text{reaching}} \quad \text{(floating)} \\ r_{\text{reaching}} + r_{\text{vertical}} + r_{\text{floating}} + r_{\text{lifting}} & \text{if } z_{\text{obj}} > \varepsilon_{\text{floating}} \quad \text{(lifting)} \end{cases} \tag{5}$$

**Jaco Push Box with Obstacle**    The reward function for task *jaco-push-box-with-obstacle* is defined as follows:

$$r_t = \begin{cases} r_{\text{reaching}} + r_{\text{vertical}} & \text{(reaching)} \\ r_{\text{reaching}} + r_{\text{vertical}} + r_{\text{pushing}} & \text{if } \texttt{dist}(\boldsymbol{p}_{\text{tcp}}, \boldsymbol{p}_{\text{obj}}) < \varepsilon_{\text{reaching}} \quad \text{(pushing)} \end{cases} \quad (6)$$

**Jaco Pick and Stack**    The reward function for task *jaco-pick-and-stack* is defined as follows:

$$r_t = \begin{cases} r_{\text{reaching}} + r_{\text{vertical}} & \text{(reaching)} \\ r_{\text{reaching}} + r_{\text{vertical}} + r_{\text{floating}} & \text{if } \texttt{dist}(\boldsymbol{p}_{\text{tcp}}, \boldsymbol{p}_{\text{obj}}) < \varepsilon_{\text{reaching}} \quad \text{(floating)} \\ r_{\text{reaching}} + r_{\text{vertical}} + r_{\text{floating}} + r_{\text{lifting}} & \text{if } z_{\text{obj}} > \varepsilon_{\text{floating}} \quad\quad\quad\quad \text{(lifting)} \end{cases} \quad (7)$$

where $r_{\text{reaching}} = \frac{1}{N} \texttt{tolerance}\big(\texttt{dist}(\boldsymbol{p}_{\text{tcp}}, \boldsymbol{p}_{\text{obj}}), \varepsilon_{\text{reaching}}\big)$, $r_{\text{pushing}} = \frac{1}{N} \texttt{tolerance}\big(\texttt{dist}(\boldsymbol{p}_{\text{target}}, \boldsymbol{p}_{\text{obj}}), \varepsilon_{\text{pushing}}\big)$, $r_{\text{vertical}} = \frac{1}{N} \texttt{tolerance}\big(\texttt{cosdist}(\boldsymbol{v}_{\text{hand}}, \boldsymbol{u}_{\text{-z}}), 1 - \varepsilon_{\text{vertical}}\big)$, $r_{\text{floating}} = \frac{1}{N} \frac{z_{\text{obj}}}{z_{\text{target}}}$, and $r_{\text{lifting}} = \frac{1}{N} \texttt{tolerance}\big(\texttt{dist}(\boldsymbol{p}_{\text{obj}}, \boldsymbol{p}_{\text{target}}), \varepsilon_{\text{lifting}}\big)$. $\boldsymbol{p}_{\text{tcp}}$ represents the position of the end-effector's center point, $\boldsymbol{p}_{\text{obj}}$ represents the position of the object (here, the box's center point), $\boldsymbol{p}_{\text{target}}$ represents the target position, $z_{\text{obj}}$ represents the $z$-axis value of the object's center point, $z_{\text{target}}$ represents the $z$-axis value of the target position, $\boldsymbol{v}_{\text{hand}}$ represents a unit vector vertical to the robot hand, and $\boldsymbol{u}_{\text{-z}}$ represents a unit vector $[0, 0, -1]$. The number of stages is represented by $N$. Here, we adopted the same function $\texttt{tolerance}$ as in DMC (Tunyasuvunakool et al., 2020).

### B.3  Task Visualizations

Figures 4 and 5 provide visualizations for behaviors generated by MERL in relation to the five 3D robotic manipulation tasks from DMC (*jaco-reach-duplo, jaco-move-box, jaco-lift-box, jaco-push-box-with-obstacle*, and *jaco-pick-and-stack*). The visualization of behaviors learned by MERL during the training procedure is shown in Figure 4. We note here that MERL solves all five of the 3D robotic manipulation tasks within 1M environment steps. This demonstrates that MERL can solve the primitive skills *reach*, *push*, *move*, *lift*, and *pick-and-place* with respect to the chosen 3D robotic manipulation tasks from DMC, within 1M environment steps. Figure 5 illustrates the visualization of successful trajectories generated by MERL.

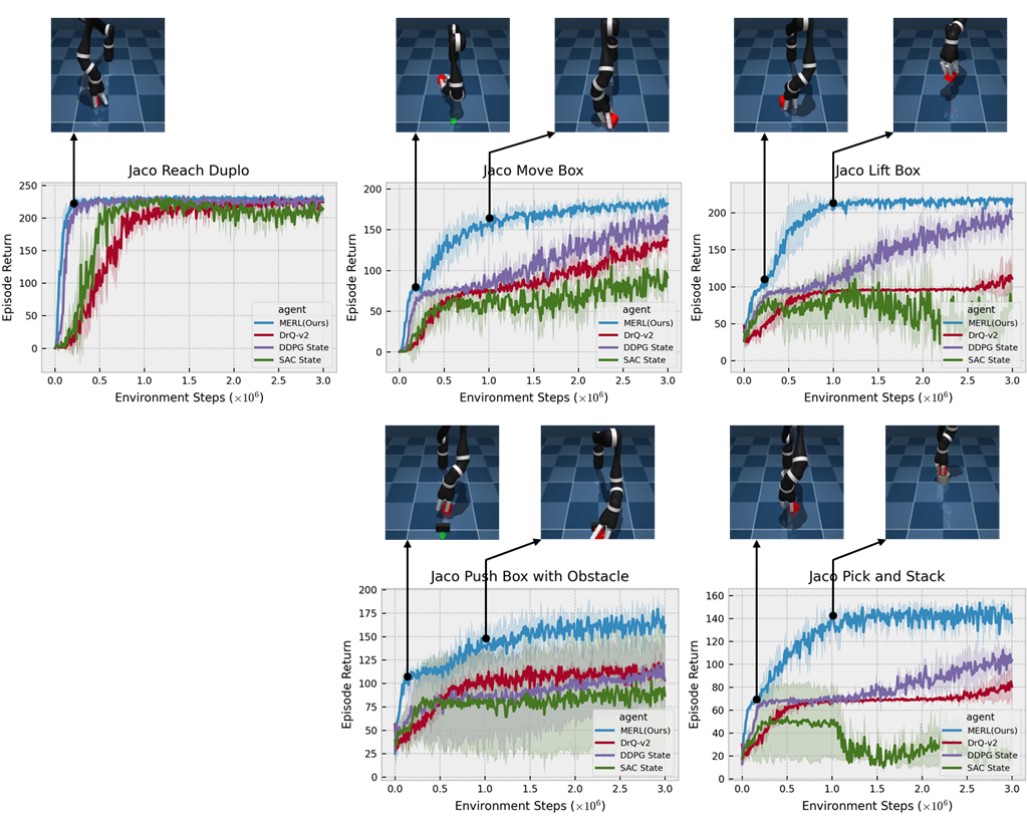

Figure 4: We visualize the behaviors learned by MERL during the training procedure in relation to the five 3D robotic manipulation tasks from DMC (*jaco-reach-duplo, jaco-move-box*, *jaco-lift-box*, *jaco-push-box-with-obstacle*, and *jaco-pick-and-stack*).

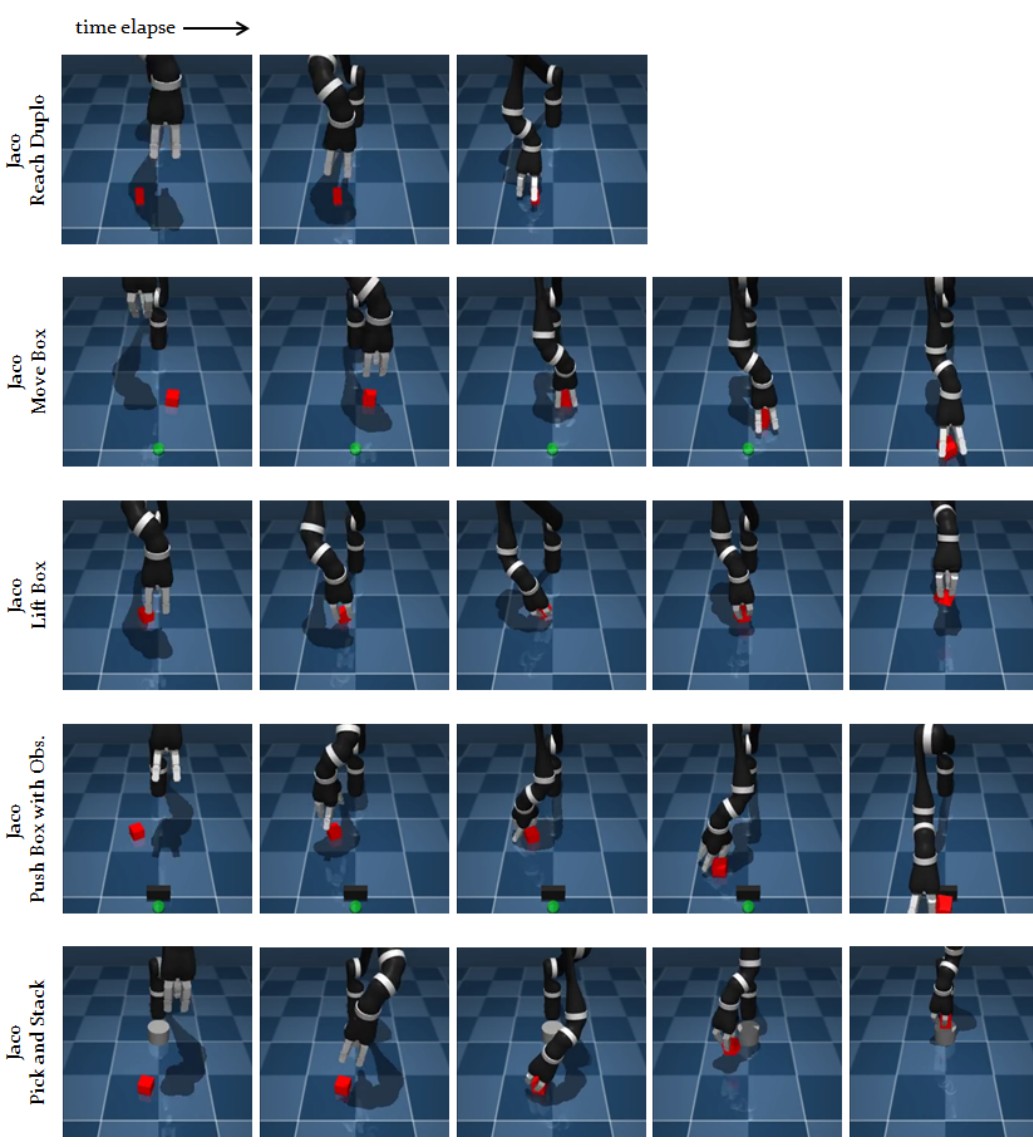

Figure 5: We visualize the successful trajectories generated by MERL in relation to the five 3D robotic manipulation tasks from DMC (*jaco-reach-duplo, jaco-move-box, jaco-lift-box, jaco-push-box-with-obstacle*, and *jaco-pick-and-stack*).

## C  IMPLEMENTATION DETAILS

In this section, we describe our implementation details for MERL. We use PyTorch as a deep learning tool and DMC and MuJoCo for our simulation. We conduct our experiments on a workstation with an Intel i9-9980XE CPU and Nvidia Quadro RTX 8000 GPU. During training, we ran five random seeds for a reliable comparison; that is, the experimental results are averaged over five different seeds.

### C.1  NETWORK ARCHITECTURES

For all networks, we initialize the weight matrix of the convolutional and fully-connected layers with an orthogonal initialization (Saxe et al., 2013) and set the bias to be zero.

**Image Encoder Network**    The image encoder network is modeled as four convolutional layers with $3{\times}3$ kernels and 32 channels, as in SAC+AE (Yarats et al., 2021c). An `ReLU` activation is applied after each convolutional layer. We use stride 1 everywhere, except for the first convolutional layer, which has stride 2. Here, we note that only the critic optimizer is allowed to update the image encoder network weights (that is, we prevent the actor's gradients from updating the image encoder network weights).

**Proprioception Encoder Network**    The proprioception encoder network is modeled as a 2-layer MLP with `ReLU` activations after each layer. Here, the final version of our method uses the proprioception encoder as the `identity` encoder. This is because as shown in Figure 6, contrary to the results in (Lee et al., 2019a; 2020b), the case where the proprioception encoder is set as the `identity` encoder provides better performance in relation to the three 3D robotic manipulation tasks from DMC (*jaco-reach-duplo, jaco-move-box*, and *jaco-lift-box*), compared to the case where the proprioception encoder is set as an MLP encoder. This suggests that the proprioception itself is enough to learn a latent multimodal representation and a policy in an efficient, joint, and end-to-end manner, without the need for additional encodings; that is, it is already well-encoded. Note that only the critic optimizer is allowed to update the proprioception encoder network weights.

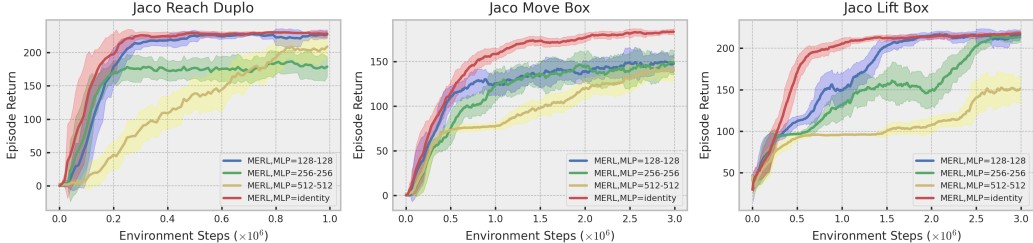

Figure 6: An additional ablation study that led us to the final version of MERL in relation to the proprioception encoder. We observe that the proprioception encoder set as the `identity` encoder (*red*) provides the best performance for the given 3D robotic manipulation tasks compared to MLP encoders (*blue, green, and yellow*).

**Multimodal Fusion Network**    The encoded visual and proprioceptive representations are fed into multimodal fusion to learn a latent multimodal representation. The multimodal fusion network is modeled as a single fully-connected layer normalized by `LayerNorm` (Ba et al., 2016). After `LayerNorm`, the `tanh` nonlinearity is applied to the output of the fully-connected layer, which serves to match the latent multimodal representation and action to the same scale. Finally, each output is concatenated to produce a single latent multimodal representation vector of dimension d. By means of the decoupled architecture, the weights of each multimodal fusion network for actor and critic are updated by the gradients of actor and critic, respectively.

**Actor and Critic Networks**    As in TD3 (Fujimoto et al., 2018), we use clipped double Q-learning for the critic network, where each Q-function is parametrized as a 3-layer (in the case of the task *jaco-reach-duplo*) or 6-layer (in the case of the tasks *jaco-move-box, jaco-lift-box, jaco-push-box-with-obstacle*, and *jaco-pick-and-stack*) MLP with `LayerNorm` and `ReLU` activations after each layer

except the last. The actor network is also modeled as a 3-layer (in the case of the task *jaco-reach-duplo*) or 6-layer (in the case of the tasks *jaco-move-box*, *jaco-lift-box*, *jaco-push-box-with-obstacle*, and *jaco-pick-and-stack*) MLP with `LayerNorm` and `ReLU` activations after each layer except the last, which outputs the mean and covariance for the diagonal Gaussian that represents the policy. The hidden dimension is set to 1024 for both the critic and the actor.

## C.2 HYPER-PARAMETERS

We provide a comprehensive overview of the hyper-parameters used in the case of the five 3D robotic manipulation tasks from DMC (*jaco-reach-duplo, jaco-move-box*, *jaco-lift-box*, *jaco-push-box-with-obstacle*, and *jaco-pick-and-stack*) in Table 1.

Table 1: An overview of hyper-parameters.

| Parameter | Setting |
| --- | --- |
| Replay buffer capacity | $10^6$ |
| Mini-batch size | 256 |
| Frame stack | 3 |
| Seed frames | 4000 |
| Exploration steps | 2000 |
| Action repeat | 1 |
| Discount factor | 0.99 |
| Optimizer | Adam |
| Learning rate | $10^{-4}$ |
| Soft-update rate | 0.01 |
| $n$-step returns | 3 |
| Exploration std. dev. | 0.2 |
| Exploration std. dev. clip | 0.3 |
| Hidden dim. for actor–critic | 1024 |
| Latent dim. for multimodal representation | 128 |

## D   DECOUPLED VS SHARED ARCHITECTURE FOR MULTIMODAL FUSION

We provide a more detailed description of the shared architecture for multimodal fusion to clarify the difference between the decoupled and shared architecture for multimodal fusion. As shown in Figure 7, in the shared architecture, a single representation is used to train both the actor network and the critic network in tandem. In contrast, the proposed decoupled architecture (see Figure 1) uses multiple representations (latent multimodal) to train the actor and critic networks separately. As a result, each of the representations can be learned to better optimize the losses of the actor and critic, respectively. More specifically, the weights of each multimodal fusion network for the actor and critic are updated by the gradients of the actor and critic, respectively. Consequently, as shown in the ablation studies, the proposed architecture leads to a remarkable performance gain in sample efficiency. To the best of our knowledge, MERL is the first RL method to apply a decoupled architecture to actor–critic learning.

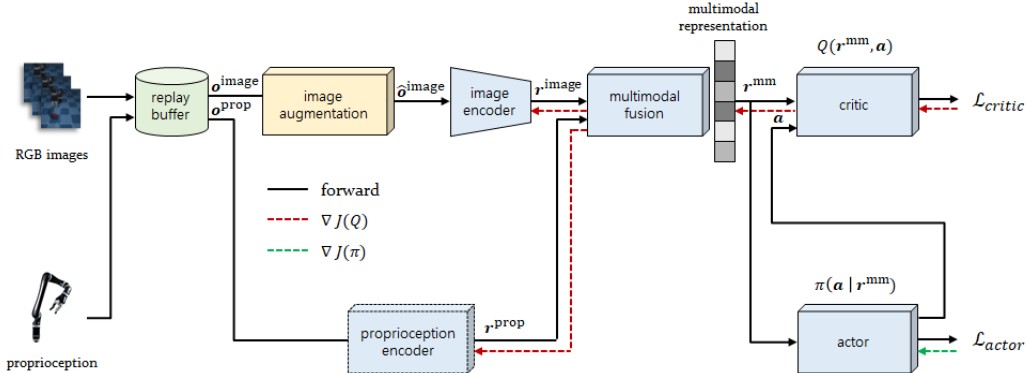

Figure 7: We provide the shared architecture for multimodal fusion. In the shared architecture, a single representation is used to train both the actor network and the critic network in tandem. Note that, in the case of the shared architecture for multimodal fusion, updating the multimodal fusion with the actor's gradients greatly hinders the agent's performance.

# E    IMPORTANCE OF LAYER NORMALIZATION IN THE STUDY OF RL

As shown in the ablation studies, we found that the use of layer normalization in multimodal fusion plays a crucial role in controlling the gradient scales and leads to a favorable sample efficiency. In light of this, we further study whether the use of layer normalization in other baselines (DrQ-v2 and DDPG State) would have a similar effect on performance. To this end, we perform additional experiments for two different cases: original baselines with and without layer normalization applied. As shown in Figure 8, the former case produces better performance in relation to the five 3D robotic manipulation tasks from DMC (*jaco-reach-duplo, jaco-move-box*, *jaco-lift-box*, *jaco-push-box-with-obstacle*, and *jaco-pick-and-stack*) than the latter case. This suggests that the use of layer normalization in RL algorithms is crucial to the study of RL. We note that the use of layer normalization in RL algorithms has received little attention in the RL community.

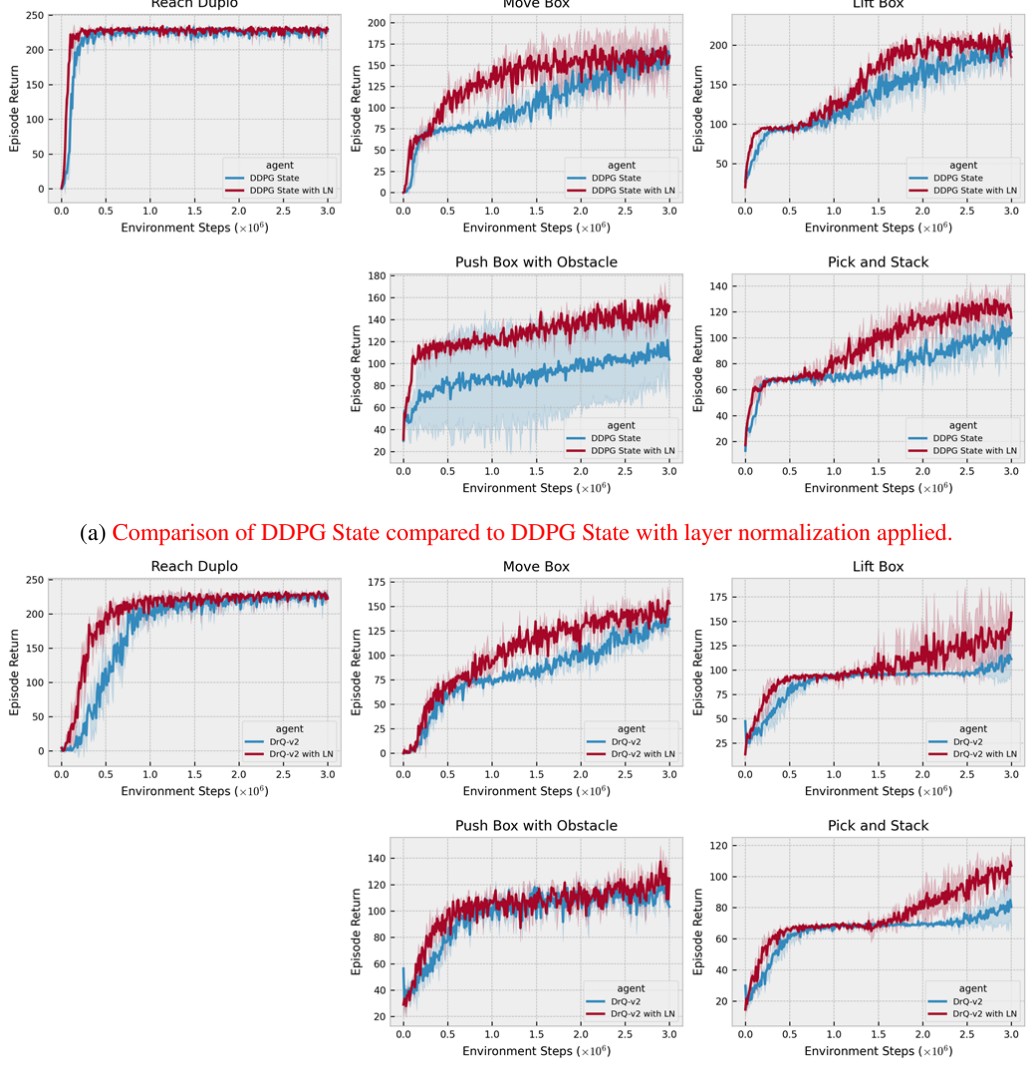

(a) Comparison of DDPG State compared to DDPG State with layer normalization applied.

(b) Comparison of DrQ-v2 compared to DrQ-v2 with layer normalization applied.

Figure 8: We compare the original baselines to those with layer normalization applied in relation to five 3D robotic manipulation tasks from DMC (*jaco-reach-duplo, jaco-move-box*, *jaco-lift-box*, *jaco-push-box-with-obstacle*, and *jaco-pick-and-stack*). We observe that the use of layer normalization in RL algorithms is crucial to the study of RL.

# F EXPERIMENTAL RESULTS FOR HUMANOID LOCOMOTION TASKS

We provide additional experimental results in relation to three complex 3D humanoid locomotion tasks (*humanoid-stand*, *humanoid-walk*, and *humanoid-run*) — these tasks being among the most difficult of 3D continuous control problems — to clarify the superiority of MERL compared to DrQ-v2. As shown in Figure 9, MERL significantly outperforms DrQ-v2 with respect to sample efficiency. Most notably, MERL solves each of the three complex 3D humanoid locomotion tasks within 5M environment steps, whereas DrQ-v2 requires 30M environment steps to solve the same tasks (see DrQ-v2 paper (Yarats et al., 2021a)). The experimental results demonstrate that MERL is one of the most effective RL methods for solving 3D continuous control problems, including robotic manipulation and locomotion tasks, with respect to learning a policy in an end-to-end manner, without the need for human-crafted representations or prior expert demonstrations.

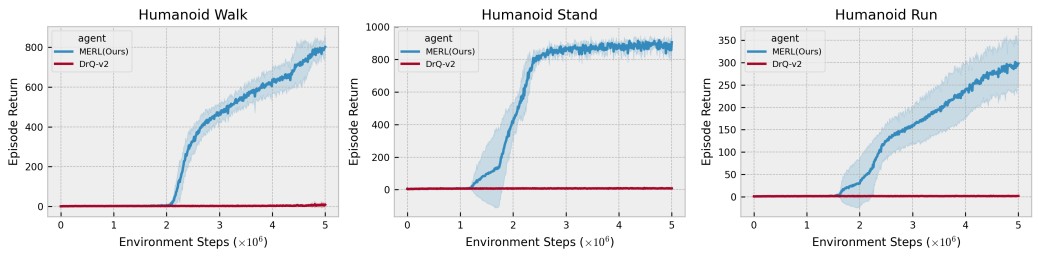

Figure 9: We compare MERL to DrQ-v2 in relation to three complex 3D humanoid locomotion tasks from DMC (*humanoid-stand*, *humanoid-walk*, and *humanoid-run*). MERL demonstrates its superiority compared to DrQ-v2, currently a state-of-the-art visual RL.

## F.1 TASK VISUALIZATIONS

Figures 10 and 11 provide visualizations for behaviors generated by MERL in relation to three complex 3D humanoid locomotion tasks from DMC (*humanoid-stand*, *humanoid-walk*, and *humanoid-run*). The visualization of behaviors learned by MERL during the training procedure is shown in Figure 10. We note that MERL solves all three of the complex 3D humanoid locomotion tasks within 5M environment steps. Figure 11 illustrates the visualization of successful trajectories generated by MERL.

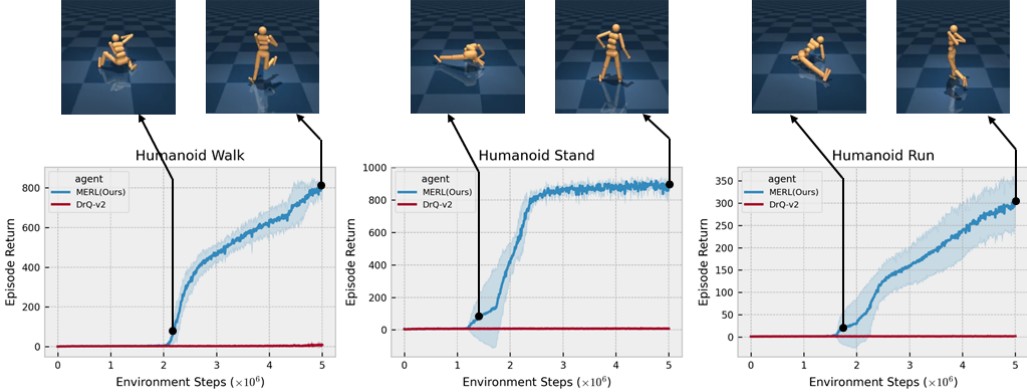

Figure 10: We visualize the behaviors learned by MERL during the training procedure in relation to three complex 3D humanoid locomotion tasks from DMC (*humanoid-stand*, *humanoid-walk*, and *humanoid-run*).

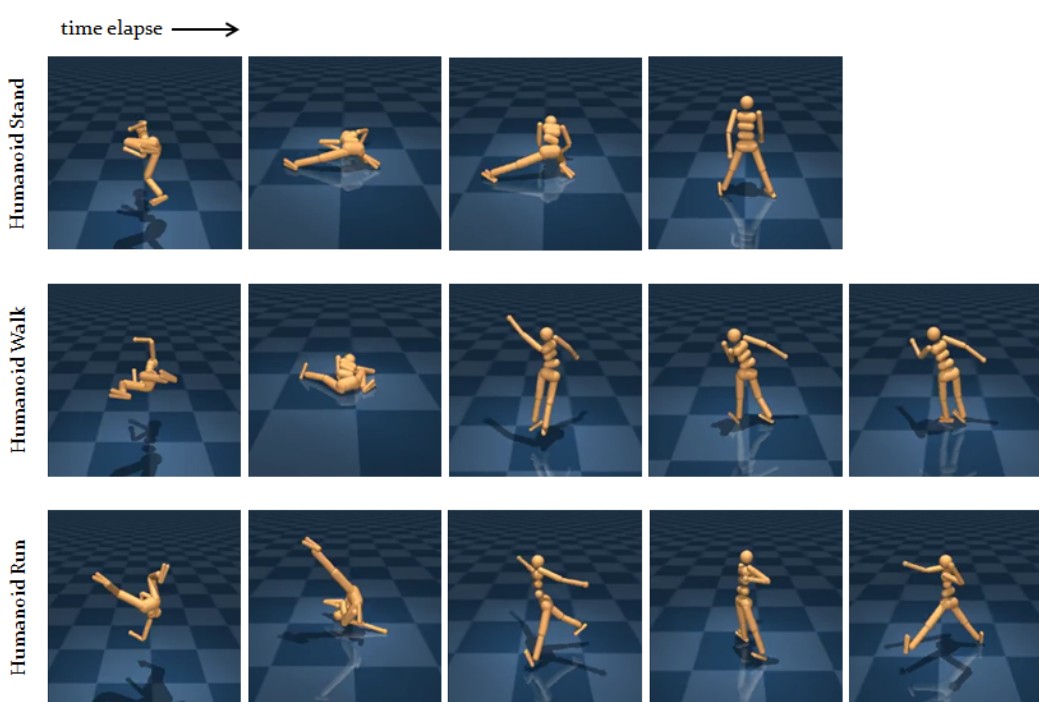

Figure 11: We visualize the successful trajectories generated by MERL in relation to three complex 3D humanoid locomotion tasks from DMC (*humanoid-stand*, *humanoid-walk*, and *humanoid-run*).

# G   COMPARISON WITH MORE VISUAL RL BASELINES

We provide the additional experimental results obtained when comparing MERL to other visual RL baselines (pixel-to-action approaches) in relation to three 3D locomotion tasks from DMC (*quadruped-walk*, *quadruped-run*, and *humanoid-stand*). The experimental results (Figure 12) show that MERL significantly outperforms all the baselines with respect to sample efficiency, learning performance, and training stability.

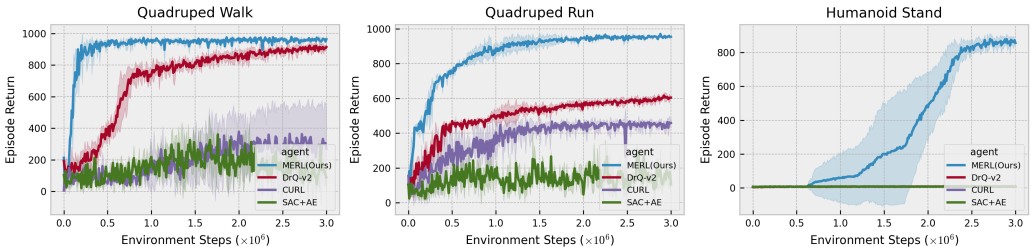

Figure 12: We compare MERL to other visual RL baselines (DrQ-v2 (Yarats et al., 2021a), CURL (Srinivas et al., 2020), and SAC+AE (Yarats et al., 2021c)) in relation to three 3D locomotion tasks from DMC (*quadruped-walk*, *quadruped-run*, and *humanoid-stand*). MERL demonstrates superior sample efficiency, significantly outperforms other visual RL baselines, and shows more stable training performance.

