# OpenReview forum: "Toward Effective Deep Reinforcement Learning for 3D Robotic Manipulation: End-to-End Learning from Multimodal Raw Sensory Data"
_ICLR.cc/2023/Conference — Submitted to ICLR 2023_

### Official Review · Reviewer_iSe9 · 2022-10-22

**Confidence:** 3
**Clarity, Quality, Novelty And Reproducibility:** Approach is clear and novel. High deg…
**Correctness:** 3
**Technical Novelty And Significance:** 4
**Empirical Novelty And Significance:** 3
**Recommendation:** 8

**Strength And Weaknesses:**

Strenghs:
- very important and broad problem to be tackling
- well-motivated, general approach
- detailed description of the algorithm, including hyperparameters
- excellent set of ablations

Weaknesses:
- Some of the prior art is somewhat misrepresented. Many of the works cited in the section which ends with 'Such methods are commonly referred to as state-based RL.' are not state-based, but learn directly end-to-end policies from pixels (Kalashnikov et al, Ibarz et al for instance). [ADDRESSED]
- The paper compares against a single baseline (DrQ-v2), but only on a narrow subset of the tasks that DrQ-v2 was able to perform. In particular, being able to train humanoid from pixels was one of the strong claims from DrQ-v2, and experiments in this setting are notably absent from this paper, and (somewhat conveniently) excluded by making this a manipulation-centric paper. This makes this reviewer very suspicious that maybe the authors tried their approach on humanoid, it didn't work well, and they maybe opted to not show these results. If that were the case, I would insist on these results to be incorporated in the paper, so that a reader would have a complete picture of the behavior of the approach. If there are other reasons why focussing on the three Jaco tasks made sense, I would love to see it being made explicit in the paper. [ADDRESSED]
- The paper does a good job at contrasting the architecture choices made against DrQ-v2. It would have been interesting to 1) further contrast with other pixel-to-action approaches that have been explored in the past, and 2) try to provide some intuition as to why this alternative architecture was interesting to consider beyond merely the empirical results. [ADDRESSED]
- No real-world results. I am very suspicious of any result which only evaluates algorithms on 'simulated pixels'. There is little evidence for a strong correlation between relative performance in these settings and real-world settings. [Not addressed, but OK to not have in scope of this paper.]
- No discussion of systems aspects: training step and inference speed, memory in contrast to other approaches. [Not addressed, but OK to not have in scope of this paper.]


**Summary Of The Paper:**

This paper presents a new reinforcement learning architecture for learning robot behaviors from pixels.

**Summary Of The Review:**

Solid contribution, however validated against a single baseline, on a narrow set of evaluations, only in simulation. To make this paper truly convincing, I would want to see 1) a broader palette of pixel-to-actions tasks being explored or 2) justification as to why only manipulation ought to be in scope, or 3) real-world results.

---

> ### Author Response · Authors · 2022-11-18
> **Author Response to Reviewer iSe9 (1/2)**
>
>
> We would like to thank you for your elaborate review and constructive feedback. We have revised our paper based on your comments, and we respond to individual points below.
>
> > Regarding the following:
> > * “Many of the works cited in the section which ends with 'Such methods are commonly referred to as state-based RL.' are not state-based, but learn directly end-to-end policies from pixels (Kalashnikov et al, Ibarz et al for instance).”
>
> To rectify this, we have amended the paragraph as follows:
> * “Deep RL has seen widespread success across a variety of domains, including board and video games and robotics and control (Mnih et al., 2015; Silver et al., 2017; Vinyals et al., 2019; Levine et al., 2016; Zhu et al., 2020; Kalashnikov et al., 2021; Ibarz et al., 2021; Kroemer et al., 2021). In recent years, a number of deep RL methods have been successfully applied to 3D robotic manipulation tasks ranging from ‘easy’ (for example, _reach-target_) to ‘hard’ (for example, _assembly_). Of these methods, some require hand-engineered components for perception, state estimation, and low-level control, for they learn from human-crafted representations (for example, (Yamada et al., 2020; Lee et al., 2019b; 2021b; Nam et al., 2022)), and such are commonly referred to as state-based RL.”
>
>
> > Regarding the following:
> > * “being able to train humanoid from pixels was one of the strong claims from DrQ-v2, and experiments in this setting are notably absent from this paper, and (somewhat conveniently) excluded by making this a manipulation-centric paper. This makes this reviewer very suspicious that maybe the authors tried their approach on humanoid, it didn't work well, and they maybe opted to not show these results.”
>
> Considering your comment, to clarify the superiority of MERL compared to DrQ-v2, we have added experimental results in relation to three complex 3D humanoid locomotion tasks (_humanoid-stand_, _humanoid-walk_, and _humanoid-run_) — these tasks being among the most difficult of 3D continuous control problems — to Appendix F. Most notably, MERL solves the three complex 3D humanoid locomotion tasks within 5M environment steps, whereas DrQ-v2 requires 30M environment steps to solve the same tasks.
>
>
> > Regarding the following:
> > * “further contrast with other pixel-to-action approaches that have been explored in the past…”
>
> Considering your comment, we have added the experimental results obtained when comparing MERL to other pixel-to-action approaches in relation to complex 3D locomotion tasks to Appendix G. The experimental results show that MERL significantly outperforms all the baselines with respect to sample efficiency, learning performance, and training stability.
>
>
> > Regarding the following:
> > * “try to provide some intuition as to why this alternative architecture was interesting to consider beyond merely the empirical results.”
>
> Some of our intuition for MERL with respect to model architecture is as follows:
> * Regarding actor–critic learning in previous RL studies, a single representation is used to train both the actor network and the critic network in tandem. We hypothesized that the performance of actor–critic learning could be improved by feeding multiple representations into the actor and critic separately and learning these representations in such a manner so as to better optimize the losses of the actor and critic, respectively. Accordingly, we applied a decoupled architecture to multimodal fusion instead of using a conventional shared architecture. Consequently, the decoupled architecture for multimodal fusion resulted in a remarkable performance gain in sample efficiency. To the best of our knowledge, MERL is the first RL method to apply a decoupled architecture to actor–critic learning.
> * In addition, we hypothesized that the use of layer normalization in multimodal fusion will play a crucial role in controlling the gradient scales and lead to a favorable sample efficiency. It turns out that the use of layer normalization did result in a significant performance gain in sample efficiency and training stability. In light of this, we proceeded to study whether the use of layer normalization in other baselines would have a similar effect on performance; indeed, we found that the use of layer normalization in RL algorithms is crucial to the study of RL. We note that the use of layer normalization in RL algorithms has received little attention in the RL community.
>
> In light of the above, we have added further details of the difference between the decoupled and shared architecture for multimodal fusion and of the importance of the use of layer normalization in the study of RL to Appendices D and E, respectively.

---

> > ### Author Response · Authors · 2022-11-18
> > **Author Response to Reviewer iSe9 (2/2)**
> >
> >
> > > Regarding the following:
> > > * “justification as to why only manipulation ought to be in scope…”
> >
> > As shown in the experimental results in relation to the three complex 3D humanoid locomotion tasks (_humanoid-stand_, _humanoid-walk_, and _humanoid-run_) in Appendix F, MERL is remarkably effective in solving overall 3D continuous control problems, including robotic manipulation and locomotion tasks. Therefore, MERL is not limited to 3D robotic manipulation, but is an effective method for overall 3D continuous control problems. Nevertheless, the reason why we targeted the 3D robotic manipulation tasks was that it is more suitable for RL study using multimodal raw sensory data (for example, RGB images and proprioception) because 3D robotic manipulation tasks require the agent to learn a policy that can control a robotic manipulator consisting of a robotic arm and hand, while simultaneously manipulating an object given in the environment.
> >
> >
> > > Regarding the following:
> > > * “No real-world results. I am very suspicious of any result which only evaluates algorithms on 'simulated pixels'. There is little evidence for a strong correlation between relative performance in these settings and real-world settings.”
> >
> > We agree with your opinion. However, real-world testing is beyond the scope of this paper. As mentioned in the submitted version of the paper, as a future work, we will seek to test and validate MERL in a real-world robotic environment through sim-to-real.
> >
> >
> > > Regarding the following:
> > > * “no discussion of systems aspects: training step and inference speed, memory in contrast to other approaches.”
> >
> > Since the baselines and MERL have been conducted in simulation, a comparative evaluation from the perspective of aspects of the system is not strictly necessary. In addition, other baselines evaluated in simulation do not provide a discussion of aspects of the system. For future work, we will discuss and perform a comparative evaluation of MERL with other studies from the perspective of aspects of the system after applying MERL to a real-world robotic environment through sim-to-real.

---

> > > ### Comment · Reviewer_iSe9 · 2022-11-29
> > > **Thank you for all the updates.**
> > >
> > > They make for a much stronger, interesting paper.
> > > I recommend acceptance, and would love to see real-world evaluations as a follow-up to this work.

---

> > > > ### Author Response · Authors · 2022-11-30
> > > > **Thank you for your reply and update**
> > > >
> > > > Dear Reviewer iSe9,
> > > >
> > > > We are very happy to hear that you have updated your recommendation to that of ‘acceptance’. Thank you for all of your comments, for they have allowed us to significantly improve our paper.
> > > >
> > > > As you suggested, we will seek to test and validate our work in a real-world robotic environment through sim-to-real.
> > > >
> > > > We sincerely thank you for your time and efforts in reviewing our paper.
> > > >
> > > > Sincerely,
> > > >
> > > > Authors

---

### Official Review · Reviewer_tRt9 · 2022-10-23

**Confidence:** 4
**Correctness:** 3
**Technical Novelty And Significance:** 1
**Empirical Novelty And Significance:** 1
**Recommendation:** 3

**Clarity, Quality, Novelty And Reproducibility:**

The clarity of the paper is good, however I think there is no novelty in it (unless I have completely missed some important thing of this work).

**Strength And Weaknesses:**

The paper is well written, clear and structured in a nice way. The method is introduced in a clear way, as well as the experimental setup.
Some quite important pieces of the approach adopted are not described in the main paper, such as the formulation of the reward stages used to train the agents, which actually can play a very important role.
The empirical evaluation includes interesting ablations of the method, examining the impact of various design choices.
Some important information are missing or unclear. What did you mean with the sentence "Unlike in ... state-based RL such as DDPG and SAC, our method stores transition ... in the replay buffer,"? Both DDPG and SAC use a replay buffer.
My main concern is that, in my opinion (unless I missed something important from the paper), there is no novelty. Vision+proprioception based agents have been trained successfully in several robotics domains, including manipulation (eg https://openreview.net/forum?id=U0Q8CrtBJxJ, http://proceedings.mlr.press/v87/kalashnikov18a.html, etc). The two modalities considered, that is proprioception and vision, have been explored in several works, and most recent works look at other challenging modalities in the context of multimodal representations especially in robot manipulation, in particular using tactile or audio signals, for example, in addition to vision and proprioception.
Other considerations, like the observation that having gradients from the agents going through the representation module during training hinders the performance, are well-known as well.
What differentiate your approach for existing vision+proprioception RL approaches, especially in the domain of robot manipulation?
Training directly from vision is in fact a challenging task, in particular it can be very data-inefficient; how did you approach this problem and what made your approach overcome it?

**Summary Of The Paper:**

The paper introduces MERL, Multimodal End-to-end Reinforcement Learning, a framework that combines multimodal (namely vision and proprioception) representation learning and RL.

**Summary Of The Review:**

The paper is well written, but lacks novelty.

---

> ### Author Response · Authors · 2022-11-18
> **Author Response to Reviewer tRt9 (1/3)**
>
> Dear Reviewer tRt9,
>
> We notice that your main concern regarding our paper is novelty. You mention that “Vision+proprioception based agents have been trained successfully in several robotics domains, including manipulation (eg [1],[2],etc).” Following this, you mention that “there is no novelty in [the paper] (unless I have completely missed some important thing of this work).”
> [1] https://openreview.net/forum?id=U0Q8CrtBJxJ
> [2] http://proceedings.mlr.press/v87/kalashnikov18a.html
>
> We believe that our work has solid contributions and novelties compared to previous studies. We first summarize the main contributions and novelties of our work below and then discuss what differentiates our work from the references that you cited; that is [1] and [2].
> * Unlike previous studies, our method, MERL, is capable of jointly learning a latent multimodal representation and a policy in an end-to-end manner from multimodal raw sensory data, without the need for human-crafted representations or prior expert demonstrations.
> * To the best of our knowledge, MERL is the “first” model-free off-policy RL method not only to learn a latent multimodal representation and a policy in an efficient, joint, and end-to-end manner from multimodal raw sensory data, but also to show a new state-of-the-art performance by significantly outperforming both current state-of-the-art visual RL and state-based RL methods with respect to sample efficiency, learning performance, and training stability.
> * Most notably, MERL is able to solve the primitive skills _reach_, _push_, _move_, _lift_, and _pick-and-place_ with respect to the chosen 3D robotic manipulation tasks from DMC, within 1M environment steps, without the need for human-crafted representations or prior expert demonstrations. Also, MERL is able to solve three complex 3D humanoid locomotion tasks (_humanoid-stand_, _humanoid-walk_, and _humanoid-run_) — these tasks being among the most difficult of 3D continuous control problems — within 5M environment steps and without the need for human-crafted representations or prior expert demonstrations. In contrast, DrQ-v2 requires 30M environment steps to solve the same three complex 3D humanoid locomotion tasks. We believe that MERL is one of the most effective RL methods for solving 3D continuous control problems, including robotic manipulation and locomotion tasks, with regards to learning a policy in an end-to-end manner, without the need for human-crafted representations or prior expert demonstrations.
> * Regarding the model architecture, we propose a novel decoupled architecture for multimodal fusion to improve the performance of actor–critic learning. Regarding actor–critic learning in previous RL studies, a single representation is used to train both the actor network and the critic network in tandem. In contrast, the proposed decoupled architecture uses multiple representations (latent multimodal) to train the actor and critic networks separately. As a result, each of the representations can be learned to better optimize the losses of the actor and critic, respectively. As shown in the ablation studies, the proposed decoupled architecture leads to a remarkable performance gain in sample efficiency. To the best of our knowledge, MERL is the first RL method to apply a decoupled architecture to actor–critic learning.
> * Also, the ablation studies show that the use of layer normalization in multimodal fusion leads to a significant performance gain in sample efficiency and training stability. Furthermore, as shown in the additional experimental results for layer normalization, which we have added to Appendix E, the use of layer normalization in RL algorithms is crucial to the study of RL. We note that the use of layer normalization in RL algorithms has received little attention in the RL community.

---

> > ### Author Response · Authors · 2022-11-18
> > **Author Response to Reviewer tRt9 (2/3)**
> >
> > Regarding [1], the authors in [1] propose an RL approach combined with a vision-based interactive policy distillation and simulation-to-reality transfer. To this end, they first train expert policies via off-policy RL in simulation with a shaped reward from states, not image pixels. Then, they use imitation learning in simulation to distill the state-based experts into a single “student” vision-based policy. What differentiates our work from [1] is as follows:
> > * In [1], the agent first learns a policy using an off-policy RL method from state, where the domain is fully observable (object position, and so on). Then, imitation learning methods, not RL methods, are used to generate a vision-based policy. By contrast, MERL is an off-policy RL method that can be trained in an end-to-end manner learning directly from multimodal raw sensory data, without the need for human-crafted representations or prior expert demonstrations. Notably, MERL is capable of learning a latent multimodal representation and a policy in an efficient, joint, and end-to-end manner from multimodal raw sensory data.
> > * In [1], there is a lack of comparison to state-of-the-art methods. By contrast, in our work, we show that MERL significantly outperforms both current state-of-the-art visual RL and state-based RL methods with respect to sample efficiency, learning performance, and training stability.
> > * To the best of our knowledge, MERL is the first model-free off-policy RL method not only to learn a latent multimodal representation and a policy in an efficient, joint, and end-to-end manner from multimodal raw sensory data, but also to show a new state-of-the-art performance by significantly outperforming both current state-of-the-art visual RL and state-based RL methods with respect to sample efficiency, learning performance, and training stability.
> >
> > Regarding [2], the authors in [2] introduce QT-Opt, a scalable self-supervised vision-based RL framework that can leverage a lot of real-world grasp attempts to train a Q-function to perform closed-loop, real-world grasping. What differentiates our work from [2] is as follows:
> > * In [2], QT-Ops learns a policy for grasping via a combination of off-policy and on-policy training, and it only generates binary gripper opening and closing commands. By contrast, MERL learns a policy for 3D robotic manipulation via off-policy RL and is capable of controlling the whole of a robotic manipulator, including a robotic arm and hand, as opposed to just a gripper.
> > * In [2], QT-Ops needs tremendous offline data to learn a policy for vision-based grasping. By contrast, MERL needs no prior data to learn a policy for 3D robotic manipulation from multimodal raw sensory data (RGB images and proprioception).
> > * Most notably, MERL is able to solve the primitive skills _reach_, _push_, _move_, _lift_, and _pick-and-place_ with respect to the chosen 3D robotic manipulation tasks from DMC, within 1M environment steps, without the need for human-crafted representations or prior expert demonstrations. Also, MERL is able to perform three complex 3D humanoid locomotion tasks (_humanoid-stand_, _humanoid-walk_, and _humanoid-run_) — these tasks being among the most difficult of 3D continuous control problems — within 5M environment steps and without the need for human-crafted representations or prior expert demonstrations. In contrast, DrQ-v2 requires 30M environment steps to solve the same three complex 3D humanoid locomotion tasks. We believe that MERL is one of the most effective RL methods for solving 3D continuous control problems, including robotic manipulation and locomotion tasks, with regards to learning a policy in an end-to-end manner, without the need for human-crafted representations or prior expert demonstrations.
> >
> > To clarify the novelty of our work, we have added discussion about the main contributions of MERL in Section 6.

---

> > > ### Author Response · Authors · 2022-11-18
> > > **Author Response to Reviewer tRt9 (3/3)**
> > >
> > >
> > > > Regarding the following:
> > > > * “Some quite important pieces of the approach adopted are not described in the main paper, such as the formulation of the reward stages used to train the agents, which actually can play a very important role.”
> > >
> > > Regrettably, we were unable to include a description of the formulation of the reward design in the main text, owing to having a strict maximum page limit of 9 pages. In light of this, we had to place the description in Appendix B.2. We agree with you on the importance of the reward function design in our RL study, but the main contributions and novelties were to take precedence.
> > >
> > >
> > > > Regarding the following:
> > > > * “Unlike in ... state-based RL such as DDPG and SAC, our method stores transition ... in the replay buffer, Both DDPG and SAC use a replay buffer.”
> > >
> > > The baselines (DrQ-v2, DDPG, and SAC), which are off-policy RL methods, make use of a replay buffer. The difference between MERL and the baselines, with respect to the use of a replay buffer, is that MERL uses two different types of raw sensory data (RGB images and proprioception). In other words, MERL uses a multimodal observation as an observation, where the multimodal observation comprises three consecutive prior RGB images and a proprioception vector.
> > >
> > >
> > > > Regarding the following:
> > > > * “Other considerations, like the observation that having gradients from the agents going through the representation module during training hinders the performance, are well-known as well.”
> > >
> > > We agree with you that the fact that updating a convolutional encoder that provides a visual representation with the actor's gradients hinders the agent's performance is well-known in visual RL. However, in this study, we propose a new decoupled architecture for multimodal fusion that uses multiple representations (latent multimodal) to train the actor and critic networks separately. In the proposed decoupled architecture, each of the representations is learned to better optimize the losses of the actor and critic, respectively. More specifically, the weights of each multimodal fusion network for actor and critic are updated by the gradients of actor and critic, respectively. As shown in the ablation studies, the proposed decoupled architecture leads to a remarkable performance gain in sample efficiency. To the best of our knowledge, MERL is the first RL method to apply a decoupled architecture to actor–critic learning.

---

> ### Author Response · Authors · 2022-12-06
> **A Gentle Reminder for Reviewer tRt9**
>
> Dear Reviewer tRt9,
>
> We sincerely appreciate your time and efforts in reviewing our paper and hope that you’ve had a chance to read our responses to your comments and the revised paper.
>
> We kindly remind you that the discussion period will end on Monday, 12th December 2022. Please, would you reply before this date and inform us of (i) whether our responses and clarifications have addressed your concerns about the novelty of our work, (ii) whether you are planning to update your recommendation, and (iii) whether you have any remaining specific concerns?
>
> If there are any outstanding issues that might prevent you from recommending ‘acceptance’, we will be more than happy to address them.
>
> Sincerely,
>
> Authors

---

> ### Author Response · Authors · 2022-12-12
> **A Gentle Reminder Again for Reviewer tRt9**
>
> Dear Reviewer tRt9,
>
> We sincerely appreciate your time and efforts in reviewing our paper and hope that you’ve had a chance to read (i) our responses to your comments, (ii) the revised paper, and (iii) our reminder.
>
> We kindly remind you again that the discussion period will end on Monday, 12th December 2022. It would be greatly appreciated if you could reply before this date and inform us of (i) whether our responses and clarifications have addressed your main concerns about the novelty of our work and (ii) whether you are planning to update your recommendation score.
>
> We believe that you may have completely missed the novelty of our work. We believe that our work has solid contributions and novelties compared to previous studies, including the references that you cited. If you have any remaining specific concerns, please do not hesitate to let us know.
>
> Sincerely,
>
> Authors

---

### Official Review · Reviewer_c2N6 · 2022-10-24

**Confidence:** 3
**Correctness:** 3
**Technical Novelty And Significance:** 2
**Empirical Novelty And Significance:** 2
**Recommendation:** 5

**Clarity, Quality, Novelty And Reproducibility:**

Clarity: The paper is quite well-written and clear. However, one point which I think would benefit from additional explanation is the ablation of the shared vs. decoupled architecture ablation study. It is not obvious how the shared architecture performs multimodal fusion; a more specific description or diagram would be helpful.

Quality: It’s not noted how many random seeds the experiments are conducted over. This would be helpful for understanding the significance of the results.

Novelty: The idea of having separate encoders for different modalities has been previously explored (Lee et. al, 2019), but not in the same setting as this work. The novelty is primarily in the application of this type of method to this particular setting.

The authors have thoroughly provided implementation details such that this work seems reproducible.


**Strength And Weaknesses:**

Strengths:
- The proposed method is conceptually simple and addresses an important problem of how to perform multimodal fusion for policy learning.
- The empirical results on the Jaco tasks are strong.
- The paper includes detailed motivation and description of the design decisions for the method, which is quite important for model-free deep RL methods.
- The paper is well-written and clear.

Weaknesses:
- Overall, I feel that the paper is too narrow in its scope for the claims that it makes. While the method is proposed to be a general multimodal framework, it is only demonstrated in a single setting with proprioceptive and image data, rather than consider other modalities like touch, sound, etc.
- Furthermore, the evaluation is only performed on three DMC tasks, and there are many specific design decisions that are made that could impact performance on these tasks in particular.
- According to the ablation study, the performance improvements of the method seem to be largely attributable to the use of layer normalization in the architecture. It would be helpful to understand if layer norm only helps MERL or if the other baselines also improve on this task with the use of layer norm.


**Summary Of The Paper:**

This paper presents an algorithm called Multimodal End-to-end Reinforcement Learning (MERL) that integrates visual and proprioceptive observations in learning model-free reinforcement learning policies. Specifically, the method builds on SAC and DrQv2, and adds encoders to handle both proprioceptive and image encoders into the policy and value function architectures. The method is evaluated on simulated 3D robotic manipulation tasks from DMC (Jaco tasks), and is compared to SAC from state and DrQv2. The authors also perform an ablation analysis of their method, in terms of how multimodal fusion is performed, the size of the multimodal representation, use of layer normalization, and exploration noise.

**Summary Of The Review:**

Overall, the paper is clear, detailed, and well-executed, but I think that in its current form, there are three main shortcomings: 1) only proprioception + vision are tested, 2) the ablation studies don’t completely demonstrate why MERL is helpful (see layer norm comment above), and 3) there are a limited number of tasks and only from the DMC suite.

---

> ### Author Response · Authors · 2022-11-18
> **Author Response to Reviewer c2N6 (1/2)**
>
> We would like to thank you for your elaborate review and constructive feedback. We have revised our paper based on your comments, and we respond to individual points below.
>
>
> > Regarding the following:
> > * “I feel that the paper is too narrow in its scope for the claims that it makes. While the method is proposed to be a general multimodal framework, it is only demonstrated in a single setting with proprioceptive and image data, rather than consider other modalities like touch, sound, etc.”
>
> We would like to thank you for pointing this out. Considering your comment, we have changed the title of our paper to more accurately reflect the content of the paper: “Toward Effective Deep Reinforcement Learning for 3D Robotic Manipulation: Multimodal End-to-End Reinforcement Learning from Visual and Proprioceptive Feedback.” Regarding other modalities such as touch and sound, we have already mentioned in Section 6 that in future work we will seek to use more sensors (for example, depth and haptic sensors) of differing modalities. Therefore, considering other modalities such as touch and sound is beyond the scope of this paper.
>
>
> > Regarding the following:
> > * “It would be helpful to understand if layer norm only helps MERL or if the other baselines also improve on this task with the use of layer norm.”
>
> We would like to thank you for your suggestion. In light of your comment, we have added the experimental results of applying layer normalization to other baselines to Appendix E to clarify the importance of the use of layer normalization in the RL study. Through the experiments, we found that the use of layer normalization in actor–critic networks as well as multimodal fusion leads to a performance gain in sample efficiency in the cases of MERL and other baselines. This suggests that the use of layer normalization in RL algorithms is crucial to the study of RL. We note that the use of layer normalization in RL algorithms has received little attention in the RL community.
>
>
> > Regarding the following:
> > * “It is not obvious how the shared architecture performs multimodal fusion; a more specific description of diagram would be helpful.”
>
> Considering your comment, we have added a more detailed description of the shared architecture for multimodal fusion to Appendix D to clarify the difference between the decoupled and shared architecture for multimodal fusion.
>
>
> > Regarding the following:
> > * “It’s not noted how many random seeds the experiments are conducted over.”
>
> We would like to thank you for pointing this out. We have added the information of how many random seeds we used for our experiments in Appendix C.

---

> > ### Author Response · Authors · 2022-11-18
> > **Author Response to Reviewer c2N6 (2/2)**
> >
> > > Regarding the following:
> > > * “there are a limited number of tasks and only from the DMC suite.”
> >
> > In light of the above comment, we have expanded the number of 3D robotic manipulation tasks to include “_jaco-push-box-with-obstacle_” and “_jaco-pick-and-stack_”, both of which are more complicated and difficult than the original three tasks (_jaco-reach-duplo_, _jaco-move-box_, and _jaco-lift-box_). The experimental results for the additional tasks show that MERL continues to outperform both current state-of-the-art visual RL and state-based RL methods with respect to sample efficiency, learning performance, and training stability. In the case of our experiments, MERL was able to solve the primitive skills _reach_, _push_, _move_, _lift_, and _pick-and-place_ with respect to the chosen 3D robotic manipulation tasks from DMC, within 1M environment steps.
> >
> > To clarify the superiority of MERL compared to DrQ-v2, we performed additional experiments in relation to three complex 3D humanoid locomotion tasks from DMC (_humanoid-stand_, _humanoid-walk_, and _humanoid-run_)—these tasks being among the most difficult of 3D continuous control problems. We have included the experimental results for the three complex 3D humanoid locomotion tasks in Appendix F. The results show that MERL significantly outperforms DrQ-v2 with respect to sample efficiency. Most notably, MERL solves each of the three complex 3D humanoid locomotion tasks within 5M environment steps, whereas DrQ-v2 solves the same tasks within 30M environment steps.
> >
> > Our experimental results demonstrate that MERL is one of the most effective RL methods for solving 3D continuous control problems, including robotic manipulation and locomotion tasks, with respect to learning a policy in an end-to-end manner, without the need for human-crafted representations or prior expert demonstrations.
> >
> > Owing to the short period for paper revisions and responses to reviewers’ comments, and hardware limitations, we were not able to perform additional experiments in other simulation environments. In light of this, we will release our full code by providing a GitHub link to the final version of our paper. In this way, RL researchers will be able to apply the code to more numerous and complex 3D continuous control problems and to test MERL in different environments. We expect experimental results in other simulation environments to be the same as those in DMC environments.

---

> > > ### Comment · Reviewer_c2N6 · 2022-12-06
> > > **Response to authors**
> > >
> > > Thank you for your detailed responses and clarifications, and your efforts in completing additional experiments and updating the paper.
> > >
> > > I appreciate the efforts that have been made to clarify the scope of the paper in terms of what kinds of modalities are considered, and I think that is now much more clear!
> > >
> > > I also think that the analysis of the contribution of layer norm to the overall method's performance is very helpful for understanding which components are the most important for achieving strong performance.
> > >
> > > I have a few remaining concerns, however:
> > > - Regarding the novelty of using a decoupled actor-critic architecture, to my understanding, there are prior works that note the interference when training actor and critic networks with shared weights, and propose to decouple the architectures to address this issue. One example is Phasic Policy Gradient (Cobbe et. al, 2020).
> > > - I appreciate the efforts to provide the comparisons for the humanoid tasks as well as the additional jaco tasks. However, one point of concern is that it may not be fair to compare the sample efficiency of the proposed method directly to Drq-v2 as Drq-v2 is not provided proprioceptive states, which may be much more helpful for the humanoid task compared to in the jaco tasks, as the humanoid task consists entirely of rearranging the proprioceptive state of the agent (there are no external objects present). For these, as well as the other tasks presented in Appendix G, it seems like it would be more fair to compare with a state-based baseline that is also provided the proprioceptive state of the humanoid.
> > > - Unfortunately due to the time constraints, non-DMC environments were still unable to be considered.
> > >
> > > I believe that the analyses presented in this paper, especially regarding layer norm, are a valuable contribution, but the main points of novelty argued in the paper are similar to those of prior works, and the experimental evaluation could benefit from more variety in domains considered.

---

> > > > ### Author Response · Authors · 2022-12-08
> > > > **Author Response to Reviewer c2N6 (1/2)**
> > > >
> > > > We thank you for your feedback on the revised paper. Please find our responses to your three main concerns below:
> > > >
> > > > > Regarding the following:
> > > > > * "Regarding the novelty of using a decoupled actor-critic architecture, to my understanding, there are prior works that note the interference when training actor and critic networks with shared weights, and propose to decouple the architectures to address this issue. One example is Phasic Policy Gradient (Cobbe et. al, 2020)."
> > > >
> > > > [1] Cobbe et. al, 2022. https://proceedings.mlr.press/v139/cobbe21a.html
> > > >
> > > > Please note that we present both a decoupled architecture for multimodal fusion (that is, decoupled representation learning for better actor–critic learning) and a decoupled actor–critic architecture, not simply a decoupled actor–critic architecture in isolation.
> > > >
> > > > We believe that our work has solid contributions and novelties regarding the decoupled architecture for multimodal fusion compared to previous studies, including the reference that you cited. What differentiates our work from [1] is as follows:
> > > > * In [1], the authors propose a Phasic Policy Gradient (PPG); that is, an RL framework that modifies conventional “on-policy” actor–critic RL methods by separating the training of policy and value networks into distinct phases. The novel contribution of [1] is the inclusion of periodic auxiliary phases to distill features from the value network into the policy network. The authors of [1] mention that whether or not to share parameters between the policy and the value networks is an important implementation decision with respect to “on-policy” actor–critic RL methods. In this context, PPG uses disjoint policy and value networks to reduce interference between objectives. Here, we should note that “off-policy” RL methods, including SAC and DDPG, use separate policy and value networks to avoid interference between their respective objectives. This is also clearly mentioned in [1].
> > > > * MERL, as an “off-policy” actor–critic RL method, also uses separate policy and value networks to avoid interference between their respective objectives. We note here that we propose a novel decoupled architecture for _multimodal fusion_ (not simply a decoupled actor–critic architecture) to further improve the performance of already separative actor–critic learning. Regarding actor–critic learning in previous RL studies, a single representation is used to train both the actor network and the critic network in tandem. In contrast, the proposed decoupled architecture for multimodal fusion uses multiple representations (latent multimodal) to train the actor and critic networks separately. As a result, each of the representations can be learned to better optimize the losses of the actor and critic, respectively. Such decoupled representation learning allows for further improvement of the performance of actor–critic learning. As shown in the ablation studies, the proposed decoupled architecture leads to a remarkable performance gain in sample efficiency compared to a conventional shared architecture (here, the conventional shared architecture shares a representation but has separative networks for the actor and critic). To the best of our knowledge, MERL is the first RL method to apply a decoupled architecture for multimodal fusion to actor–critic learning.

---

> > > > > ### Author Response · Authors · 2022-12-08
> > > > > **Author Response to Reviewer c2N6 (2/2)**
> > > > >
> > > > > > Regarding the following:
> > > > > > * However, one point of concern is that it may not be fair to compare the sample efficiency of the proposed method directly to Drq-v2 as Drq-v2 is not provided proprioceptive states, which may be much more helpful for the humanoid task compared to in the jaco tasks, as the humanoid task consists entirely of rearranging the proprioceptive state of the agent (there are no external objects present). For these, as well as the other tasks presented in Appendix G, it seems like it would be more fair to compare with a state-based baseline that is also provided the proprioceptive state of the humanoid."
> > > > >
> > > > > We agree with you that it would be unfair to compare our method, MERL, with visual RL methods, including DrQ-v2, with respect to sample efficiency in relation to 3D locomotion tasks that do not require the manipulation of an external object in the environment. Considering your comment, we have performed additional experiments for a state-based baseline, DDPG State, in relation to three complex 3D humanoid locomotion tasks (_humanoid-stand_, _humanoid-walk_, and _humanoid-run_) to perform a fair comparative evaluation.
> > > > >
> > > > > The experimental results show that DDPG State provides better sample efficiency performance than MERL in relation to the tasks _humanoid-stand_ and _humanoid-walk_, whereas MERL provides better sample efficiency performance than DDPG State in relation to _humanoid-run_. Such results suggest the following: (1) in the case of 3D locomotion tasks that do not require the manipulation of an external object in the environment, the use of proprioceptive feedback is sufficient for the learning of a policy and (2) multimodal feedback, including image pixels, is helpful in the learning of a policy, even in the case of tasks that require only an agent’s internal state when such tasks become progressively more difficult. Here, the experimental results of comparison between MERL and both DrQ-v2 and DDPG State in relation to the three complex 3D humanoid locomotion tasks (_humanoid-stand_, _humanoid-walk_, and _humanoid-run_) is available at https://www.dropbox.com/s/xuttis8rk1i7nyt/results_humanoid_tasks.png?dl=0. We will add such results to the final version of our paper at a later date, for we are currently unable to revise our paper during the discussion stage.
> > > > >
> > > > >
> > > > > > Regarding the following:
> > > > > > * Unfortunately due to the time constraints, non-DMC environments were still unable to be considered."
> > > > >
> > > > > Considering your comment, we are currently working on applying MERL to a simulation environment (for example, meta-world [2]) that is different to that of the DMC manipulation environment. We are currently working on the modification of an observation space and the input/output structure of a replay buffer to enable MERL to use multimodal inputs for the robotic manipulation tasks of meta-world. We expect experimental results in other simulation environments based on the MuJoCo physics simulation to be the same as those in the DMC manipulation environment. This is because the success of a task depends more on task complexity and reward function design than on a specific simulation environment. Here, task complexity includes the degrees of freedom of a robotic manipulator, whether to use a gripper or a robotic hand, whether a task is long-horizon or not, and features of the robot environment (for example, the number, size, or shape of external objects). If tasks given in other simulation environments have similar complexity and appropriately designed reward functions to those given in the DMC manipulation environment, MERL will show the same experimental results as those presented in this study in the case of such tasks and other environments. Here, the DMC manipulation environment uses a Jaco robot comprising a robotic hand, not a gripper.
> > > > > [2] meta-world. https://meta-world.github.io/
> > > > >
> > > > > In addition, as previously mentioned, we will release our full code by providing a GitHub link to the final version of our paper. In this way, RL researchers will be able to apply the code to more numerous 3D continuous control problems, including robotic manipulation and locomotion tasks, and to test and validate MERL in numerous different simulation environments.

---

> ### Author Response · Authors · 2022-12-12
> **Thank you for your further feedback**
>
> Dear Reviewer c2N6,
>
> We sincerely appreciate your time and efforts in reviewing the revised paper and hope that you've had a chance to read our responses to your further comments.
>
> We kindly remind you again that the discussion period will end on Monday, 12th December 2022. It would be greatly appreciated if you could reply before this date and inform us of (i) whether our responses and clarifications have addressed your main concerns and (ii) whether you are planning to update your recommendation score.
>
> We believe that you may have misunderstood the novelty of a decoupled architecture for multimodal fusion. If you have any remaining specific concerns, please do not hesitate to let us know.
>
> Thank you once again for your thorough reviews and insighful comments.
>
> Sincerely,
>
> Authors

---

### Official Review · Reviewer_gi8i · 2022-10-27

**Confidence:** 4
**Correctness:** 3
**Technical Novelty And Significance:** 2
**Empirical Novelty And Significance:** 2
**Recommendation:** 5

**Clarity, Quality, Novelty And Reproducibility:**

The paper is well written and easy to follow. Its novelty is marginal. Reproducibility is conditional on authors open-sourcing the code and authors don't mention this possibility.

**Strength And Weaknesses:**

The paper present results which are beyond the state of the art. However, presented results are limited to three relatively simple tasks for which a staged reward was needed. The proposed approach is relatively standard and the major theoretical contribution seems to be just a marginal improvement with respect to existing alternatives.

**Summary Of The Paper:**

The paper presents a method to end-to-end learn manipulation tasks from raw multi-modal sensory data. Multimodality refers to the use of both propriocetion and RGB-images. The proposed RL method is model-free and off-policy. It relies on image augmentation to learn a multimodal representations fed to an actor and a critic network. The proposed methodology is tested on three robot manipulation tasks from the Deepmind Control suite. Comparisons with state-of-the-art alternatives (i.e. DrQ-v2, DDPG from state and SAC from state). Remarkably the proposed approach does better than all other approaches, including DDPG from state. Ablations studies show: (1) the importance of decoupled representations for the actor and the critic networks, (2) the importance of choosing the representation dimensions, (3) the importance of layer normalization and (4) the minor benefits of scheduled exploration noise.

**Summary Of The Review:**

I think the paper is relatively well executed and the published results are a significant step with respect to the state of the art. However, nothing really new is presented in the paper to the point that it becomes surprising to notice that these significant improvements could be achieved with minor adjustments to the network architecture (see ablation studies). Also, the paper is limited to 3 simple tasks which limits a lot the soundness of the proposed approach.

I usually try to be more verbose and provide some useful feedback in my reviews but I don't have much to say on this specific paper. My final score is "below threshold" but I am struggling with finding the right evaluation; it would be great if the community could have the code and test the approach on more numerous and more complicated manipulation tasks.

---

> ### Author Response · Authors · 2022-11-18
> **Author Response to Reviewer gi8i (1/2)**
>
> We would like to thank you for your elaborate review and constructive feedback. We have revised our paper on the basis of your comments, and we respond to individual points below.
>
> > Regarding the following:
> > * "However, presented results are limited to three relatively simple tasks for which a staged reward was needed."
> > * "Also, the paper is limited to 3 simple tasks which limits a lot the soundness of the proposed approach."
>
> In light of the above comments, we have expanded the number of 3D robotic manipulation tasks to include “_jaco-push-box-with-obstacle_” and “_jaco-pick-and-stack_”, both of which are more complicated and difficult than the original three tasks (_jaco-reach-duplo_, _jaco-move-box_, and _jaco-lift-box_). The experimental results for the additional tasks show that MERL continues to outperform both current state-of-the-art visual RL and state-based RL methods with respect to sample efficiency, learning performance, and training stability. In the case of our experiments, MERL was able to solve the primitive skills _reach_, _push_, _move_, _lift_, and _pick-and-place_ with respect to the chosen 3D robotic manipulation tasks from DMC, within 1M environment steps.
>
> To clarify the superiority of MERL compared to DrQ-v2, we performed additional experiments in relation to three complex 3D humanoid locomotion tasks from DMC (_humanoid-stand_, _humanoid-walk_, and _humanoid-run_)—these tasks being among the most difficult of 3D continuous control problems. We have included the experimental results for the three complex 3D humanoid locomotion tasks in Appendix F. The results show that MERL significantly outperforms DrQ-v2 with respect to sample efficiency. Most notably, MERL solves each of the three complex 3D humanoid locomotion tasks within 5M environment steps, whereas DrQ-v2 solves the same tasks within 30M environment steps.
>
> Our experimental results demonstrate that MERL is one of the most effective RL methods for solving 3D continuous control problems, including robotic manipulation and locomotion tasks, with respect to learning a policy in an end-to-end manner, without the need for human-crafted representations or prior expert demonstrations.

---

> > ### Author Response · Authors · 2022-11-18
> > **Author Response to Reviewer gi8i (2/2)**
> >
> > > Regarding the following:
> > > * “The proposed approach is relatively standard and the major theoretical contribution seems to be just a marginal improvement with respect to existing alternatives.”
> > > * “nothing really new is presented in the paper to the point that it becomes surprising to notice that these significant improvements could be achieved with minor adjustments to the network architecture (see ablation studies).”
> >
> > To the best of our knowledge, MERL is the “first” model-free off-policy RL method not only to learn a latent multimodal representation and a policy in an efficient, joint, and end-to-end manner from multimodal raw sensory data without the need for human-crafted representations or prior expert demonstrations, but also to show a new state-of-the-art performance by significantly outperforming both current state-of-the-art visual RL and state-based RL methods with respect to sample efficiency, learning performance, and training stability. Most notably, MERL is able to solve the primitive skills _reach_, _push_, _move_, _lift_, and _pick-and-place_ with respect to the chosen 3D robotic manipulation tasks from DMC, within 1M environment steps. Also, it is able to solve three complex 3D humanoid locomotion tasks (_humanoid-stand_, _humanoid-walk_, and _humanoid-run_) within 5M environment steps. Such results demonstrate that MERL is currently the most efficient method for learning a policy in an end-to-end manner from multimodal raw sensory data, without the need for human-crafted representations or prior expert demonstrations.
> >
> > MERL’s state-of-the-art performance comes from a careful configuration of multimodal representation learning combined with data-augmented RL. Specifically, we propose a novel decoupled architecture for multimodal fusion to improve the performance of actor–critic learning. Regarding actor–critic learning in previous RL studies, a single representation is used to train both the actor network and the critic network in tandem. In contrast, the proposed decoupled architecture uses multiple representations (latent multimodal) to train the actor and critic networks separately. As a result, each of the representations can be learned to better optimize the losses of the actor and critic, respectively. As shown in the ablation studies, the proposed decoupled architecture leads to a remarkable performance gain in sample efficiency. To the best of our knowledge, MERL is the first RL method to apply a decoupled architecture to actor–critic learning.
> >
> > Also, the ablation studies show that the use of layer normalization in multimodal fusion leads to a significant performance gain in sample efficiency and training stability. Furthermore, as shown in the additional experimental results for layer normalization, which we have added to Appendix E, the use of layer normalization in RL algorithms is crucial to the study of RL. We note that the use of layer normalization in RL algorithms has received little attention in the RL community. We believe that MERL can contribute to the RL community in many ways (for example, expert data in offline RL), for many RL researchers can easily apply MERL to their research owing to its relative simplicity and remarkable effectiveness.
> >
> >
> > > Regarding the following:
> > > * “it would be great if the community could have the code and test the approach on more numerous and more complicated manipulation tasks.”
> >
> > In light of the above comment, we will release the full code by providing a GitHub link to the final version of our paper. In this way, RL researchers will be able to apply the code to more numerous and complex 3D continuous control problems, including robotic manipulation and locomotion tasks.

---

> ### Author Response · Authors · 2022-12-12
> **A Gentle Reminder for Reviewer gi8i**
>
> Dear Reviewer gi8i,
>
> We sincerely appreciate your time and efforts in reviewing our paper and hope that you’ve had a chance to read (i) our responses to your comments, (ii) the revised paper, and (iii) our reminder.
>
> We kindly remind you again that the discussion period will end on Monday, 12th December 2022. It would be greatly appreciated if you could reply before this date and inform us of (i) whether our responses and clarifications have addressed your concerns and (ii) whether you are planning to update your recommendation score.
>
> Sincerely,
>
> Authors

---

### Author Response · Authors · 2022-11-18
**Summary of Revisions**

We, the authors, are sincerely grateful to the reviewers for their thorough reviews, which has helped in improving our paper. We have revised our paper accordingly (highlighted in red in the PDF), and we provide a summary of our revisions below. Also, we provide a more detailed discussion in the comment boxes of each reviewer.

#### Regarding experiments,
* [gi8i, c2N6] We added experimental results in relation to two additional difficult manipulation tasks (_jaco-push-box-with-obstacle_ and _jaco-pick-and-stack_) to Section 5 to clarify the soundness of our method, MERL.
* [gi8i, c2N6, iSe9] We added experimental results in relation to three complex 3D humanoid locomotion tasks (_humanoid-stand_, _humanoid-walk_, and _humanoid-run_) to Appendix F to clarify the superiority of MERL compared to DrQ-v2.
* [c2N6] We added experimental results in relation to the application of layer normalization to other baselines to Appendix E to verify the importance of the use of layer normalization in the study of RL.
* [iSe9] We added experimental results for other baselines based on a pixel-to-action approach to Appendix G.

#### Regarding writing,
* [gi8i, tRt9] We added discussion on the main contributions of MERL to Section 6 to clarify the novelty of our method.
* [c2N6] We changed the title of our paper to accurately reflect the contents of our paper to “Toward Effective Deep Reinforcement Learning for 3D Robotic Manipulation: Multimodal End-to-End Reinforcement Learning from Visual and Proprioceptive Feedback.”
* [c2N6] We added a more detailed description of the shared architecture for multimodal fusion to Appendix D to clarify the difference between a decoupled and shared architecture for multimodal fusion.
* [c2N6] We added information relating to the number of random seeds used in our experiments to Appendix C.
* [iSe9] We modified Section 2.1 to clarify which references relate specifically to state-based RL.

#### Regarding source code,
* [gi8i] We will release the full code by providing a GitHub link to the final version of our paper in order that RL researchers can apply it to their own research and also use and test it on more numerous and complex 3D continuous control problems, including robotic manipulation and locomotion tasks.

We hope our revisions and responses have addressed the reviewers’ concerns, and again we are sincerely grateful for their help in improving the paper. Please let us know if you have any further comments.

---

### Author Response · Authors · 2022-12-06
**A Gentle Reminder**

Dear Reviewers,

We sincerely appreciate your time and efforts in reviewing our paper and hope that you’ve had a chance to read our responses to your comments and the revised paper.

We kindly remind you that the discussion period will end on Monday, 12th December 2022. Please, would you reply before this date and inform us of (i) whether our responses and clarifications have addressed your concerns, (ii) whether you are planning to update your recommendations, and (iii) whether there are any remaining specific concerns?

We believe that we have addressed your main concerns by means of the results of supporting experiments. If you have any further concerns or questions, please do not hesitate to let us know.

We are eagerly looking forward to your feedback on the revised paper.

Thank you once again for your thorough reviews and insightful comments.

Sincerely,

Authors

---

### Decision · Program_Chairs · 2023-01-20

**Decision:**

Reject

**Justification For Why Not Higher Score:**

Novelty is unclear. Experiments are quite narrow on DMC tasks,

**Justification For Why Not Lower Score:**

N/A

**Metareview: Summary, Strengths And Weaknesses:**

The paper presents a new RL approach that combines visual and proprioceptive observation to learn policies. The reviewers appreciate the strong experimental results and comparisons with DrQv2 along with the ablation analysis. The multimodal fusion idea is conceptually simple and easy to understand. However, there is a major concern shared by the reviewers on the novelty of the work. The paper is quite narrow in scope as design decisions in 3 DMC tasks may not translate to real-world improvements on robots. The author feedback was helpful to get additional perspective. For example the 3 Humanoid tasks improved the scope. Nevertheless during the discussion phase, the reviewers found that the novelty concern still present. In the words of one of the reviewers : "I find it difficult to identify the true novelty in it. Maybe the novelty is the use of a decoupled representation, one for the actor and one for the critic? In any case, why (what is the reason behind) the method outperforming the other variants is still unclear to me."

I highly encourage the authors to strengthen the writing of the paper vis-a-vis the motivation and novelty of the method. Perhaps extending the experimental evaluation would also convince the community that the algorithm generalizes beyond the existing tasks.